# Superconductivity in an extreme strange metal

D. H. Nguyen[1], A. Sidorenko[1], M. Taupin[1], G. Knebel [2], G. Lapertot [2], E. Schuberth[3] & S. Paschen [1✉]

Some of the highest-transition-temperature superconductors across various materials classes exhibit linear-in-temperature 'strange metal' or 'Planckian' electrical resistivities in their normal state. It is thus believed by many that this behavior holds the key to unlock the secrets of high-temperature superconductivity. However, these materials typically display complex phase diagrams governed by various competing energy scales, making an unambiguous identification of the physics at play difficult. Here we use electrical resistivity measurements into the micro-Kelvin regime to discover superconductivity condensing out of an extreme strange metal state—with linear resistivity over 3.5 orders of magnitude in temperature. We propose that the Cooper pairing is mediated by the modes associated with a recently evidenced dynamical charge localization–delocalization transition, a mechanism that may well be pertinent also in other strange metal superconductors.

[1] Institute of Solid State Physics, Vienna University of Technology, Wiedner Hauptstr. 8-10, Vienna, Austria. [2] Université Grenoble Alpes, CEA, Grenoble INP, IRIG, PHELIQS, Grenoble, France. [3] Technische Universität München, Garching, Germany. ✉email: paschen@ifp.tuwien.ac.at

The phenomenon of superconductivity has fascinated scientists since its discovery in 1911. It had to await microscopic understanding, achieved in the BCS theory[1], for almost 50 years. Conventional superconductors such as aluminum and niobium are now key players in the race for realizing the quantum computer[2]. Understanding superconductivity that goes beyond this framework, as first seen in the heavy fermion compound $CeCu_2Si_2$[3] and since then considered for numerous materials classes[4], is the next grand challenge.

In several of these "unconventional" superconductors—across high-$T_c$ cuprates[5], iron pnictides and organic conductors[6], heavy fermion metals[7,8], and, very recently, infinite-layer nickelates[9], twisted bilayer graphene[10], and $WSe_2$[11]—the normal state shows a linear-in-temperature "strange metal" electrical resistivity, at least in certain temperature ranges. This suggests that to decipher this type of superconductivity requires understanding the mechanism of the underlying strange metal normal state. However, because of the complexity of the phase diagrams of many of these systems, this has proven challenging. Multiple competing effects[4], crossovers between different scaling behaviors[12], possible trivial linear-in-temperature resistivity contributions[13], or simply the fact that strong superconductivity covers much of the normal-state phase space and needs to be suppressed by external parameters[14], which may modify the original normal state, are inhibiting consensus on the key mechanism at play.

On the other hand, there is a material—the heavy fermion compound $YbRh_2Si_2$[15]—where such complications are absent and linear-in-temperature strange metal behavior[15–18] has recently been pinned down as resulting from a dynamical electron localization–delocalization transition[19] as the Kondo effect is destroyed at a magnetic quantum critical point (QCP)[20]. Alas, this QCP appeared to lack superconductivity.

Our work unblocks this situation. By developing electrical resistivity measurements down to record low temperatures—more than 1.5 orders of magnitude below state-of-the-art—and using them to study $YbRh_2Si_2$, we discover unconventional superconductivity condensing out of a further expanded range of strange metal behavior, now covering 3.5 orders of magnitude in temperature. This establishes the connection between electron localization–delocalization-derived strange metal behavior and superconductivity, discussed previously for several other materials[21–24], to a new level of confidence, thereby pointing to its universality and putting the spotlight on Cooper pairing mediated by the critical modes that are associated with this transition.

We note that in purely itinerant systems at the border of an antiferromagnetic phase with spin density wave order, antiferromagnetic paramagnons[25] may lead to deviations from Fermi liquid behavior (although generally not to strictly linear-in-temperature resistivities) and provide superconducting pairing, a mechanism evoked for $CePd_2Si_2$[26]. However, for the above strange metals[5–11] this mechanism seems unlikely, because no magnetic phase exists nearby and/or because there is evidence that this (weak-coupling) magnetic order parameter description is inappropriate.

The heavy fermion compound $YbRh_2Si_2$ exhibits a low-lying antiferromagnetic phase that is continuously suppressed to zero by a small magnetic field, establishing a field-induced QCP[16]. A linear temperature dependence of the electrical resistivity is seen below about 10 K[16–18]. This behavior is ruled out to be due to electron-phonon scattering because the non-$f$ reference material $LuRh_2Si_2$ is a normal metal and because Fermi liquid behavior is recovered when $YbRh_2Si_2$ is tuned away from the QCP by magnetic field[16–18]. The recent observation of quantum critical energy-over-temperature ($\omega/T$) scaling in the charge dynamics[19], together with jumps in the extrapolated zero-temperature Hall

coefficient[27,28] and associated thermodynamic[29] and spectroscopic signatures[30], identifies a dynamical electron localization–delocalization transition as underlying the strange metal behavior.

## Results and discussion

We have developed, and implemented in a nuclear demagnetization cryostat[31], a setup for high-resolution electrical resistivity measurements to temperatures well below 1 mK (see Methods), and used it to measure state-of-the-art $YbRh_2Si_2$ single crystals[17,32]. To assess the previously discussed[33] role of Yb nuclear moments, in addition to samples with natural abundance Yb (containing 14.2% $^{171}Yb$ with a nuclear moment $I = 1/2$ and 16.1% $^{173}Yb$ with a nuclear moment $I = 5/2$)[32], we also studied $^{174}YbRh_2Si_2$, which is free of nuclear Yb moments[17]. In zero magnetic field, both samples show the characteristic kink near the Néel temperature $T_N$, as well as linear-in-temperature behavior above and Fermi liquid $T^2$ behavior below (Fig. 1a, b). The parameters extracted from these fits (Table 1) are in good agreement with previous results[15–18], confirming the high reproducibility of the properties in state-of-the-art $YbRh_2Si_2$ single crystals. At the lowest temperatures, a pronounced drop of the resistivity signals the onset of superconductivity. It is fully displayed in Fig. 1c, d. In $YbRh_2Si_2$, the transition is rather sharp ($\Delta T_c/T_c = 0.10$ for a resistance change from 90% to 10% of the value above the transition), with an onset somewhat below 9 mK. In $^{174}YbRh_2Si_2$, the onset is shifted to below 6 mK, and the transition is broadened, even though the residual resistance ratio of this sample is almost twice that of the normal $YbRh_2Si_2$ sample (Table 1), an effect that will be discussed later.

Application of magnetic fields within the $a$-$a$ plane of the tetragonal crystal successively suppresses superconductivity in both samples (Fig. 1c, d). Note, however, that clear signs of superconductivity are visible even at the quantum critical field (see Fig. 1c, green curve, for $YbRh_2Si_2$, and Fig. 2b for $^{174}YbRh_2Si_2$), thus demonstrating superconductivity that nucleates directly out of the strange metal state (Fig. 2a, b). Our measurements expand the previously established strange metal regime to a record span of linear electrical resistivity over 3.5 orders of magnitude in temperature, with a high accuracy of 5% in the linear exponent $\epsilon$ (Fig. 2c).

To characterize the superconductivity further, we performed isothermal magnetic field sweeps. In $YbRh_2Si_2$, all traces of superconductivity disappear only in fields beyond 70 mT, which is well above the quantum critical field of 60 mT (Fig. 3a). The unusual two-step-like shape of the resistivity isotherms below 5 mK, which is distinct from normal broadening in applied fields, suggests that two different superconducting phases might be involved. This is corroborated by the results on $^{174}YbRh_2Si_2$, where the lowest-temperature isotherms show clear signs of reentrance (Fig. 3b). We note that no current dependence was observed, ruling out that flux-flow resistivity is at the origin of these characteristics. Next, we present color-coded temperature–magnetic field phase diagrams of $YbRh_2Si_2$ and $^{174}YbRh_2Si_2$ (Fig. 3c, d), constructed from a large number of isotherms. Their merit is to give a general and fully unbiased impression of the phases present: an "S-shaped" superconducting region in $YbRh_2Si_2$ and two possibly separated superconducting regions in $^{174}YbRh_2Si_2$.

For more quantitative phase diagrams, a definition of the (field-dependent) transition temperatures $T_c$ and (temperature-dependent) upper critical fields $B_{c2}$ has to be adopted. We choose the midpoints of the resistive transitions, as sketched in Fig. 1c, d. For $YbRh_2Si_2$ this leads to the phase boundary delineated by the full ($B_{c2}$) and open ($T_c$) blue circles in Fig. 4a, which closely

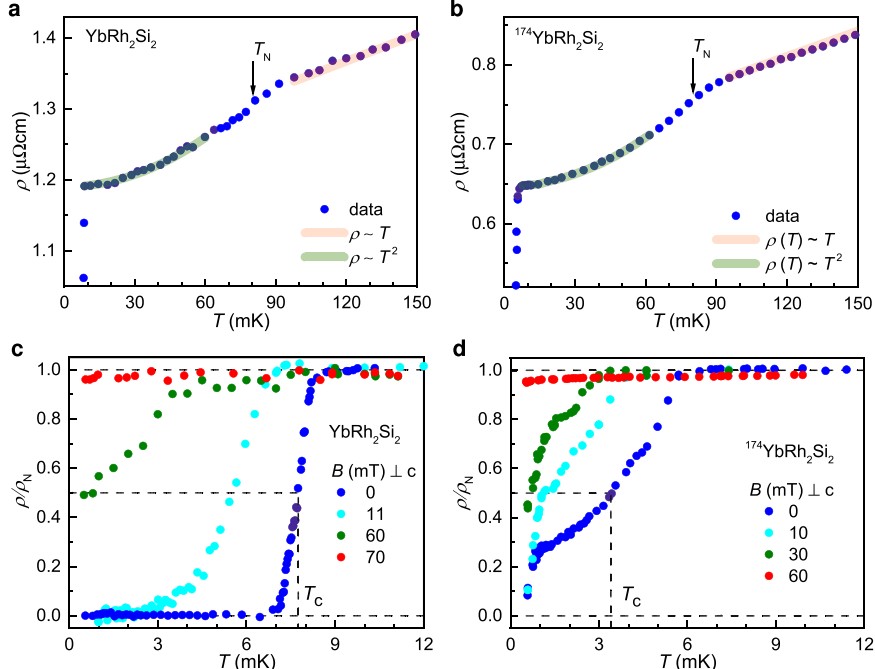

**Fig. 1 Temperature dependence of the electrical resistivity of YbRh$_2$Si$_2$ and $^{174}$YbRh$_2$Si$_2$. a, b** Electrical resistivity $\rho(T)$ below 150 mK, showing linear-in-$T$ behavior above the Neel temperature $T_N$, $T^2$ behavior below it, and the onset of superconductivity at the lowest temperatures. **c, d** Electrical resistivity below 12 mK, scaled to its normal-state resistivity $\rho_N$ just above the transition, showing the superconducting transition at $T_c$, which we define as the temperature where $\rho(T)$ has dropped to $\rho_N/2$. Magnetic fields (applied within the tetragonal $a$-$a$ plane) successively suppress $T_c$. The 10 and 30 mT curves for $^{174}$YbRh$_2$Si$_2$ were extracted from isothermal field sweeps (Fig. 3b); all other curves were recorder as function of temperature.

**Table 1 Characteristics of the investigated YbRh$_2$Si$_2$ and $^{174}$YbRh$_2$Si$_2$ single crystals.**

| Sample | Batch | RRR$_{10\,mK}$ | $\rho_0$ ($\mu\Omega$cm) | $A$ ($\mu\Omega$cm/K$^2$) | $\rho_0'$ ($\mu\Omega$cm) | $A'$ ($\mu\Omega$cm/K) | $\gamma_{KW}^{0\,T}$ (J/molK$^2$) |
|---|---|---|---|---|---|---|---|
| YbRh$_2$Si$_2$ | 63113_1 | 67 | 1.19 | 20.2 | 1.23 | 1.17 | 1.42 |
| $^{174}$YbRh$_2$Si$_2$ | Lap0288 | 123 | 0.55 | 14.8 | 0.59 | 0.85 | 1.22 |

Both samples are from batches studied in detail previously[17,32]. Their residual resistance ratios RRR$_{10\,mK}$ = R(300 K)/R(10 mK), as well as the zero-field Fermi liquid behavior $\rho = \rho_0 + AT^2$ below $T_N$ and the non-Fermi liquid behavior $\rho = \rho_0' + A'T$ at the quantum critical field of 60 mT confirm high sample quality. To remove uncertainties in the geometric factors, we have assumed $\rho(300\,K) = 80$ $\mu\Omega$cm[15]. The Sommerfeld coefficient in zero field $\gamma_{KW}^{0\,T}$, calculated from $A$ via the universal Kadowaki–Woods ratio $A/\gamma^2 = 10^{-5}$ $\mu\Omega$cm(mol K)$^2$/(mJ)$^2$, is a good estimate of the non-quantum critical contribution (see Supplementary Note 2: Estimates on Planckian dissipation).

resembles the shape drawn by the yellow color code (50% resistance) in Fig. 3c. An initial rapid suppression of $T_c$ is followed by a much more gradual one, indicating that moving toward the QCP boosts the superconductivity against the general trend of field suppression associated with the Pauli- and/or orbital-limiting effect of the magnetic field (for cartoons of this field effect, see Supplementary Fig. 3). This is seen even more clearly in the 90% resistance line (boundary of the pale shading in Fig. 4a) that exhibits a local maximum at a magnetic field only slightly below the QCP. This evidences that at least a component of the superconductivity of YbRh$_2$Si$_2$ is promoted by the same quantum critical fluctuations that are also responsible for the extreme strange metal behavior—thus anchoring both phenomena to the material's QCP. The fact that there might indeed be two distinct superconducting phases, one more readily suppressed by magnetic field and one that is less field sensitive, receives further support from the phase diagram of $^{174}$YbRh$_2$Si$_2$, presented next.

Because the resistive transitions have finite widths, they interfere if two or more phase boundaries are nearby. For $^{174}$YbRh$_2$Si$_2$, where the "unbiased" color-coded phase diagram already suggests two adjacent phases, we used a simple model to disentangle their effects (see Supplementary Note 1: Analysis of resistivity vs magnetic field isotherms and Supplementary Fig. 1). Indeed, by fitting this model to the data we find two distinct

phases, a low-field one that we denote as phase I, and a field-induced one that we call phase II (Fig. 4b, see caption for the meaning of the different symbols). Again, we also show the 90% resistance line as the boundary of the pale shading. Using this criterion, phase I and II of $^{174}$YbRh$_2$Si$_2$ grow together into a single superconducting region, similar to what is observed for YbRh$_2$Si$_2$. Conversely, this adds evidence to the above-proposed two-phase interpretation of the peculiarly shaped superconducting region of YbRh$_2$Si$_2$ (see cartoons in Supplementary Fig. 2). Despite the qualitative similarities between the phase diagrams of YbRh$_2$Si$_2$ and $^{174}$YbRh$_2$Si$_2$, it is clear that quantitatively, the superconductivity is much weaker in $^{174}$YbRh$_2$Si$_2$. Thus, whereas nuclear moments—present in YbRh$_2$Si$_2$ but absent in $^{174}$YbRh$_2$Si$_2$—are not a necessary ingredient to create superconductivity, they do considerably strengthen it.

In what follows we give a few simple estimates of characteristics of the superconductivity in YbRh$_2$Si$_2$ and $^{174}$YbRh$_2$Si$_2$ (see Table 2). From the zero-field $T_c$ values (7.9 and 3.4 mK for YbRh$_2$Si$_2$ and $^{174}$YbRh$_2$Si$_2$, respectively) and upper critical field slopes $-dB_{c2}/dT|_{T_c}$ (4.4 and 2.1 T/K, much larger than in conventional superconductors), which we determined from linear fits in Fig. 4 (see red lines), we estimate the weak-coupling BCS Ginzburg–Landau coherence lengths $\xi_{GL}$ (97 and 215 nm). Together with the relevant (non-quantum critical) Sommerfeld

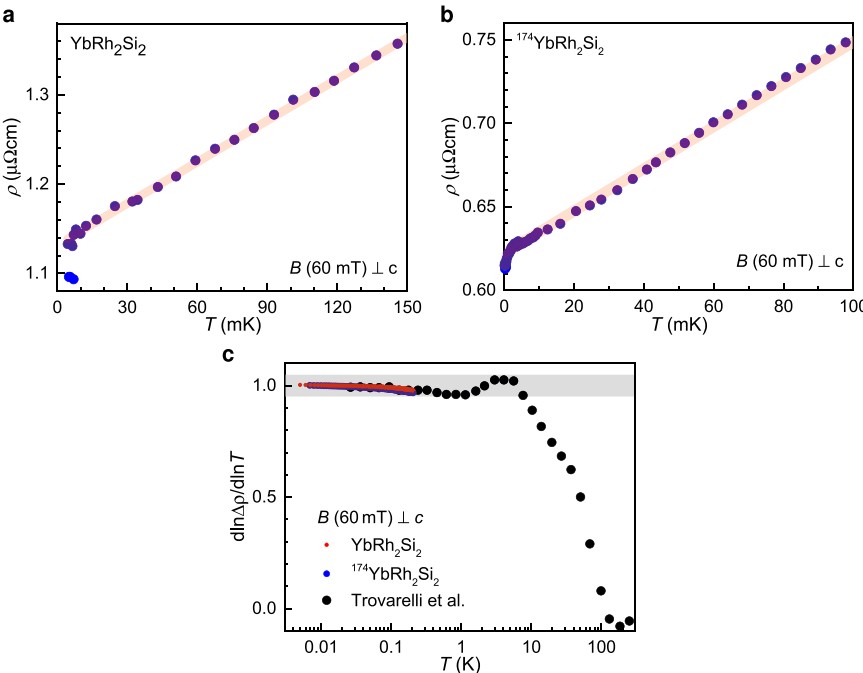

**Fig. 2 Strange metal behavior of YbRh$_2$Si$_2$ and $^{174}$YbRh$_2$Si$_2$. a, b** Temperature-dependent electrical resistivity at the quantum critical field of 60 mT applied within the tetragonal *a-a* plane. Superconductivity develops out of a strange metal $\rho \sim T$ normal state. The parameters of the linear-in-*T* fits are given in Table 1. **c** Electrical resistivity exponent $\epsilon$ of the non-Fermi liquid form $\rho = \rho'_0 + A'T^\epsilon$, determined as $\epsilon = d\ln \Delta\rho / d\ln T$, shown together with literature data up to high temperatures[15]. This establishes linear-in-*T* resistivity, evidenced by $\epsilon = 1 \pm 0.05$, over 3.5 orders of magnitude in temperature. Note that this presentation visualizes the temperature dependence of the exponent with great sensitivity, and that the error bar of ±0.05 corresponds to a high precision in the exponent's closeness to the value of 1 (for comparison see, e.g., Fig. 12 of ref. [8]).

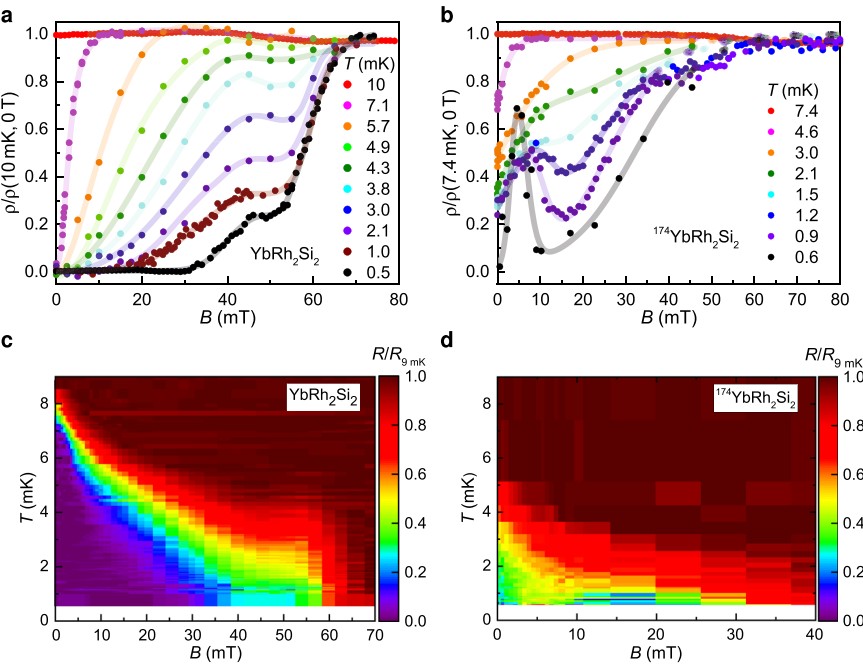

**Fig. 3 Magnetic field dependence of the electrical resistivity of YbRh$_2$Si$_2$ and $^{174}$YbRh$_2$Si$_2$. a, b** Selected isotherms of the electrical resistivity $\rho(B)$, revealing reentrant superconductivity at low temperatures in $^{174}$YbRh$_2$Si$_2$ (**b**) and remnants thereof in YbRh$_2$Si$_2$ (**a**). Lines are guides-to-the-eyes in (**a**) and fits to a multi-transition model (Supplementary Note 1: Analysis of resistivity vs magnetic field isotherms and Supplementary Fig. 1) in (**b**). **c, d** Temperature-magnetic field phase diagrams of the two samples, with the reduced electrical resistance $R/R$(9 mK) as color code. The yellow color corresponds to the resistive midpoint. As a measure of $T_c(B)$ it indicates a phase boundary of unusual shape for YbRh$_2$Si$_2$, and two possibly distinct phases for $^{174}$YbRh$_2$Si$_2$.

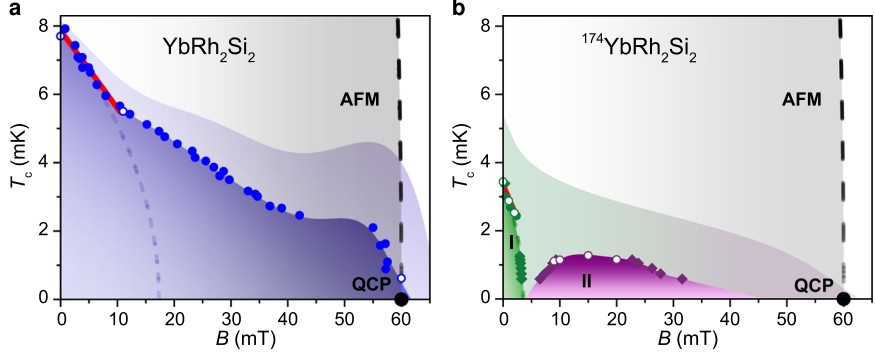

**Fig. 4 Temperature–magnetic field phase diagrams of YbRh$_2$Si$_2$ and $^{174}$YbRh$_2$Si$_2$. a** Results for YbRh$_2$Si$_2$. **b** Results for $^{174}$YbRh$_2$Si$_2$. Open ($T_c$) and full ($B_{c2}$) circles of all colors represent midpoints (50%) of the resistive transitions of iso-$B$ $\rho(T)$ curves (Fig. 1c, d) T iso-$T$ $\rho(B)$ curves (Fig. 2a, b), respectively. Full diamonds represent the $B_{c2}$ values extracted from a multi-transition model (Supplementary Note 1: Analysis of resistivity vs magnetic field isotherms and Supplementary Fig. 1) for the three involved transitions (green for leaving phase I, violet for entering and leaving phase II). The good agreement between diamonds and circles in the regions of minimal multi-phase interference justifies this procedure. The upper boundaries of the pale shaded areas away from data points represent 90% resistance lines. The black dashed line in both panels indicates the boundary of the antiferromagnetic (AFM) phase. It represents an extrapolation of data from refs. [28,29], to $T = 0$, the quantum critical point (QCP, black dot). The red lines are linear fits to the low-field (50%) data, and determine both the upper critical field slopes $-dB_{c2}/dT|_{T_c}$ and the zero-field superconducting transition temperatures $T_c$ [$=T_c(H = 0)$]. The dashed lines represent (inverted) $B'_{c2}(T) = B'_{c2}(T = 0) \cdot [1 - (T/T_c)^2]$ mean-field curves determined from $-dB_{c2}/dT|_{T_c}$ and $T_c$, as estimates of the (low-field) phase boundaries. Below 0.5 mK, all other phase/shading boundaries correspond to linear-in-$B$ extrapolations of fits to the lowest $T$ data points. Note that the perfect overlap of the full and open symbols demonstrates the high reproducibility of the results.

**Table 2 Characteristics of the superconducting phase(s) in YbRh$_2$Si$_2$ and $^{174}$YbRh$_2$Si$_2$.**

| Sample | $T_c$ (mK) | $\frac{-dB_{c2}}{dT}\|_{T_c}$ ($\frac{T}{K}$) | $\xi_{GL}$ (nm) | $k_F$ ($\frac{1}{nm}$) | $l_{tr}$ (nm) | $\lambda_{GL}$ ($\mu$m) | $B'_{c2}$ (mT) | $B_p$ (mT) | $\alpha$ |
|---|---|---|---|---|---|---|---|---|---|
| YbRh$_2$Si$_2$ | 7.9 | 4.4 | 97 | 5.2 | 371 | 1.8 | 24 | 15 | 0.011 |
| $^{174}$YbRh$_2$Si$_2$ | 3.4 | 2.1 | 215 | 4.8 | 976 | 2.0 | 5.0 | 6.4 | 0.0065 |

Both $T_c(B)$ and $B_{c2}(T)$ are defined as the midpoints of the resistive transitions (see Figs 1c, d and 2a, b). The zero-field values $T_c$ [$=T_c(H = 0)$] as well as the upper critical field slopes $-dB_{c2}/dT|_{T_c}$ are determined from linear fits to the data at small fields (red lines in Fig. 4). Listed also are estimates of the Ginzburg–Landau coherence length $\xi_{GL}$, the average Fermi wavevector $k_F$, the transport scattering length $l_{tr}$, the Ginzburg–Landau penetration depth $\lambda_{GL}$, the (orbital limiting) upper critical field $B'_{c2}$, and the Pauli limiting field $B_p$ (see text). $\alpha$ is the prefactor of the scattering rate $1/\tau = \alpha k_B T/\hbar$ of a simple Drude conductor, estimated from $T_c$, $\xi_{GL}$ (both below), and $\gamma^{0T}_{KW}$ (from Table 1), as explained in the Supplementary Note 2: Estimates on Planckian dissipation; "Planckian dissipation" refers to $\alpha = 1$.

coefficients (see Table 1) we derive the average Fermi wave-vectors $k_F$ (5.2 and 4.8 nm$^{-1}$) and, with the residual resistivities $\rho_0$ (see Table 1), the transport scattering lengths $l_{tr}$ (371 and 976 nm) and the Ginzburg–Landau penetration depths $\lambda_{GL}$ (1.8 and 2.0 $\mu$m). These describe moderately clean ($l_{tr} > \xi_{GL}$), strongly type-II ($\lambda_{GL}/\xi_{GL} \gg 1/\sqrt{2}$) superconductivity. Estimates of the orbital and Pauli limiting upper critical fields, via $B'_{c2} = -0.7 T_c dB_{c2}/dT|_{T_c}$ and $B_p = 1.764 k_B T_c/(\sqrt{2}\mu_B)$ (24 and 15 mT for YbRh$_2$Si$_2$, and 5 and 6.4 mT for $^{174}$YbRh$_2$Si$_2$, respectively), might well be compatible with the phase boundary of phase I in $^{174}$YbRh$_2$Si$_2$ and a putative corresponding low-field phase in YbRh$_2$Si$_2$. It is clear, however, that superconductivity in both YbRh$_2$Si$_2$ and $^{174}$YbRh$_2$Si$_2$ extends to much larger fields (Fig. 4), providing further evidence for the unconventional nature of the observed superconductivity.

Next we discuss how our results relate to previous thermodynamic measurements on YbRh$_2$Si$_2$ with natural abundance Yb, which provided evidence for superconductivity away from the QCP: shielding signals were detected, with onsets somewhat below 10 mK and near 2 mK, in fields up to 0.055 and 0.418 mT, respectively[33], leaving it open whether this superconductivity is related to the strange metal state of YbRh$_2$Si$_2$. The transition temperature we determine from zero-field electrical resistivity measurements on YbRh$_2$Si$_2$ (with an onset near 9 mK) is in good agreement with the upper transition (into the B phase) detected there, identifying it thus with our phase I. The observation of the lower transition (into the A + sc phase of ref. [33]) could then be taken as evidence that the two superconducting phases of YbRh$_2$Si$_2$ postulated above intersect (see

sketch in Supplementary Fig. 2a). Alternatively, our low-field superconducting region could comprise both the B and sc phase of ref. [33]; then our high-field region would be a separate phase not detected in ref. [33] (see sketch in Supplementary Fig. 2b), just as the high-field phase of $^{174}$YbRh$_2$Si$_2$. This should be clarified by future magnetization/susceptibility measurements in lower fields (below the background field of 0.012 mT reached in ref. [33], which appears to be well above the lower critical field of the B phase), ideally on powdered samples to better assess the Meissner volume of the B phase.

The lower transition of ref. [33] is accompanied by a specific heat anomaly that releases a giant amount of entropy. It was interpreted as phase transition into a state of hybrid nuclear and electronic spin order of Yb, which reduces the internal (staggered) magnetic field of the primary electronic order and thus creates a less hostile environment for superconductivity. As our study reveals, the superconductivity in $^{174}$YbRh$_2$Si$_2$, which lacks nuclear Yb moments and can thus not exhibit such hybrid order, is indeed considerably weaker than in YbRh$_2$Si$_2$. This confirms that nuclear Yb moments boost the superconductivity in YbRh$_2$Si$_2$. The fact that this mechanism apparently works up to temperatures well above the hybrid ordering temperature may be ascribed to short-range fluctuations, evidenced by the entropy release extending up to about 10 mK. Apart from the overall weakening of the superconductivity in $^{174}$YbRh$_2$Si$_2$ compared to YbRh$_2$Si$_2$, the phase diagrams of the two materials do, however, share many similarities (Fig. 4), calling for an understanding within the same framework.

The question that then arises is what is the mechanism of the Cooper pairing in the detected superconducting phases?

We start by recapitulating our results that make the BCS mechanism extremely unlikely: (i) superconductivity in YbRh$_2$Si$_2$ condenses out of an extreme strange metal state, with linear-in-temperature resistivity right down to the onset of super-conductivity (Fig. 2a); (ii) the upper critical field slope (Table 2) as well as the directly measured critical field (Fig. 4a) strongly overshoot both the Pauli and the orbital limiting fields; (iii) the low-temperature resistivity isotherms exhibit a two-step transition (Fig. 3a), evidencing that one component is much less field sensitive than the other; (iv) the superconducting phase boundary deviates strongly from a mean-field shape (Figs 3a and 4a), evidencing that the field boosts (at least part of) the super-conductivity against the general trend of field suppression; (v) superconductivity is strongly suppressed by substituting the natural abundance Yb (of atomic mass 173.04) by $^{174}$Yb, though the isotope effect in a BCS picture would have a minimal effect (a reduction of $T_c$ by 0.1%). It is thus natural to assume that quantum critical fluctuations are involved in stabilizing (at least the high-field part of) the superconductivity in YbRh$_2$Si$_2$.

Very recently, in a two-impurity Anderson model that features Kondo destruction[20], the singlet-pairing susceptibility was found to be strongly enhanced by critical local moment fluctuations[34]. Because singlet pairing may be subject to Pauli limiting, this phase—though stabilized by quantum critical fluctuations—might be suppressed by the applied magnetic field even well before the QCP is reached. Thus, phase I of $^{174}$YbRh$_2$Si$_2$ and the putative corresponding low-field phase of YbRh$_2$Si$_2$ are promising candidates for this type of superconductivity. Interestingly, in this context, unconventional superconductivity is also discussed in the spin-triplet channel[35,36]. It is tempting to consider phase II of $^{174}$YbRh$_2$Si$_2$ and the putative corresponding high-field phase of YbRh$_2$Si$_2$ to be of this kind, which would provide a compelling link to recently discovered candidate spin-triplet topological superconductors[37,38]. Of course, any model for superconductivity in YbRh$_2$Si$_2$ (and $^{174}$YbRh$_2$Si$_2$) should also correctly capture the normal state properties.

At the QCP of YbRh$_2$Si$_2$, a dynamical electron localization–delocalization transition, featuring both linear-in-temperature dc resistivity, and linear-in-frequency and linear-in-temperature "optical resistivity", was evidenced by quantum critical frequency-over-temperature scaling, with a critical exponent of 1, of the THz conductivity of YbRh$_2$Si$_2$[19]. This evidences critical modes in addition to the ones associated with the suppression of the (antiferro)magnetic order parameter. A microscopic mechanism compatible with this scaling is the disentanglement of the (electronic) Yb 4$f$ moments from the conduction electrons as static Kondo screening breaks down at the QCP[19,20,27–30]. Whether spin-triplet superconductivity may arise in models that capture this physics remains to be clarified by future work. Given that the quantum critical magnetic field is considerably larger than the scale associated with the superconducting transition temperature near the QCP, a promising direction is to consider the role of the applied magnetic field as reducing the spin symmetry from being in-plane continuous to Ising-anisotropic. In fact, calculations in an Ising-anisotropic two-impurity Bose-Fermi Anderson model suggest that near the model's Kondo-destruction QCP the spin-triplet pairing correlation is competitive with the spin-singlet one[39]. Naturally, the proposal of spin-triplet superconductivity should also be scrutinized by future experiments, including probes of anisotropies and NMR investigations, which are in principle feasible at ultralow temperatures.

Our observation of unconventional superconductivity condensing out of the textbook strange metal state of YbRh$_2$Si$_2$ ends a debate about reasons for its (previously supposed) absence. Thus, neither the 4$f$ moment stemming from Yb or the tuning parameter being magnetic field (as opposed to the typical situation of Ce-based systems under pressure or doping tuning), nor the QCP being governed by effects beyond order parameter fluctuations inhibits superconductivity. Instead, we propose that critical modes associated with a dynamical electron localization–delocalization transition mediate the strange-metal unconventional superconductivity in YbRh$_2$Si$_2$. Future experiments, ideally in conjunction with ab initio-based theoretical work, shall ascertain this assignment, disentangle the different superconducting phases, determine the symmetry of the order parameter, clarify further important details such as the single vs multiband nature of the superconductivity, and even explore the possibility of exotic surface phases.

Finally, we relate our discovery to strange-metal unconventional superconductors in other materials classes. It has been pointed out that the strange metal behavior in many of these is compatible with "Planckian dissipation", i.e., with the transport scattering rate $1/\tau$ in a simple Drude conductor being equal to $\alpha k_B T/\hbar$, with $\alpha \approx 1$[40]. For YbRh$_2$Si$_2$ and $^{174}$YbRh$_2$Si$_2$, we estimate much smaller $\alpha$ values (see Table 2, and Supplementary Note 2: Estimates on Planckian dissipation), suggesting that linear-in-temperature resistivity—even in the extreme form observed here—does not require the Planckian limit to be reached. Whether this is related to the very strongly correlated nature of this compound (with effective masses above $1000 \cdot m_0$, much beyond those in the materials considered in ref. [40]) or its pronounced deviation from Drude-like behavior (see Supplementary Materials of ref. [19]) is an interesting question for future studies. As to a microscopic understanding of the phenomenon, we point out that charge delocalization transitions, similar to those observed in YbRh$_2$Si$_2$, are being discussed also in other strange-metal superconductors, including the high-$T_c$ cuprates[21,22], an organic conductor[23] and, tentatively, even magic angle bilayer graphene[24]. Our results thus point to the exciting possibility that a dynamical electron localization–delocalization transition may mediate strange-metal unconventional superconductivity in a broad range of materials classes.

## Methods

**Refrigerator and thermometry**. All measurements were carried out in the Vienna nuclear demagnetization refrigerator[31]. We use resistance thermometry (Pt resistance thermometers Pt-1000, RuO$_2$ thermometers, and sliced Speer carbon resistance thermometers) for temperatures above 10 mK and magnetic thermometry (CMN, pulsed Pt-NMR thermometers, via the pulsed nuclear magnetic resonance of $^{195}$Pt nuclei) for temperature below 20 mK. For the presented results, we used a CMN thermometer between 20 and 2 mK, a Speer thermometer above, and a Pt-NMR thermometer below.

**Filters**. To attenuate radiofrequency radiation, a series of filters and thermalization stages consisting of thermo-coax cables (from room temperature to the 50 mK plate), silver-epoxy filters, and RC filters were installed. The silver-epoxy filters are also used for thermalization (down to the mixing chamber temperature).

**Sample holders and thermalization**. Machined silver sample holders made from a 5N silver rod were annealed to reach a residual resistance ratio of 2000. They are directly screwed to the nuclear stage using home-made silver screws. To thermalize the samples, one of the two outer contact points (of the 4-point technique, see point E below) was connected by 50 μm gold wires to the silver holder, via spot welding on the sample and screwing to the silver holder. A separate grounding point of excellent quality (anchored 600 m below ground level) was used. The different stages of the cryostat (4 and 1 K plate, still, 50 mK plate, mixing chamber, copper nuclear stage) were all connected via NbTi/CuNi superconducting wires to the same ground, such that the ground potential was highly uniform. To avoid ground loops, all measurements devices were connected via opto-couplers. During the measurements at ultralow temperatures (below 10 mK), all measurement devices were powered by $4 \times 12$ V − 150 A batteries with floating ground. For electrical isolation (away from the ground point), Vespel was used on the sample holder, the superconducting coil, and the Nb-superconducting shield.

**Magnetic field applied to the samples**. A dc magnetic field coil made of superconducting NbTi wire ($T_c = 9.2$ K), a superconducting Nb cylinder (20 mm diameter, 10 cm length), and the sample holder are concentrically assembled. The field coil and the superconducting shield are thermalized to the mixing chamber,

the sample holder to the nuclear stage. Magnetic fields at the position of the samples up to 80 mT were generated by applying a dc current, with the precision of a few µT. For the highest fields, a current above 1.4 A was passed through the self-made coil. Because this represented a considerable risk of quenching and breaking the magnet, only a final set of experiments was done up to the highest fields. In particular, most of the measurements on $^{174}$YbRh$_2$Si$_2$ were done up to 45 mT only.

**Resistivity measurements**. Electrical resistivity measurements were done with a 7124 Precision Lock-in Amplifier. Electrical currents were sourced by a CS580 voltage-controlled current source. The lowest measurement current in our experiments was 10 nA. To increase the signal to noise ratio, low-temperature transformers (encapsulated in a lead shield) with a gain of 1000 were used. They were installed at and thermalized to the mixing chamber. In addition, a SR560 low noise voltage pre-amplifier was used at room temperature. Electrical contacts for these measurements were made by spot-welding gold wires to the samples, in a standard 4-point geometry. Carefully derived measurement protocols (electrical current densities, sweep rates, waiting times, etc.) for a good thermalization of the samples to the nuclear stage were followed for all presented measurements; the good thermalization is confirmed by the reproducibility of the results between different types of experiments (e.g., temperature vs magnetic-field sweeps, cooling vs warming curves). Applying larger currents leads to overheating effects at the lowest temperatures, but no evidence for flux-flow resistance could be revealed.

## Data availability
The datasets generated during and/or analyzed during the current study are available from the corresponding author on reasonable request.

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

## Acknowledgements
The authors thank A. Casey, Ch. Enss, G. Frossatti, L. Levitin, J. Saunders, A. de Waard, D. Zumbuehl, and other colleagues from the European Microkelvin Platform (EMP) for sharing expertise in ultralow-temperature techniques, C. Krellner for growing YbRh$_2$Si$_2$ single crystals under supervision of Ch. Geibel and F. Steglich in Dresden, M. Brando for assistance in the selection of suitable single crystals, and P. Buehler, A. Prokofiev, Q. Si, and F. Steglich for fruitful discussions. The team in Vienna acknowledges financial support from the Austrian Science Fund (FWF grants P29296-N27 and DK W1243), the EMP (H2020 Project 824109), and the European Research Council (ERC Advanced Grant 227378).

## Author contributions
S.P. initiated and led the study. G.K. and G.L. synthesized and characterized the $^{174}$YbRh$_2$Si$_2$ single crystals. D.H.N. and A.S. set up the ultralow-temperature experiments, with initial guidance by E.S., and performed the measurements. D.H.N., M.T., and S.P. analyzed the data. S.P. wrote the manuscript, with input from all authors. All authors contributed to the discussion.

## Competing interests
The authors declare no competing interests.
