## [Peer Review File · Nature Communications]

Reviewers' Comments:

Reviewer #1:

Remarks to the Author:

I have read the paper by Nguyen et al. with a great deal of interest. The interplay of Planckian transport in strange metals and associated "high-temperature" superconductivity has gotten a great deal of attention in recent years, especially after Ref. 40 of the present manuscript pointed out the surprising universality associated with Drude scattering rates across microscopically distinct materials.

The field-tuned transport data presented in this manuscript is absolutely striking. The data presented in Figs. 2 and 3, especially the near-perfect T-linear behavior of the resistivity down to such low temperatures is remarkable and certainly deserves to be published. (I am not up to date on the latest experimental literature regarding YRS. I am assuming that the data presented here down to such low temperatures doesn't exist elsewhere in the literature).

However, I have a few questions that the authors might want to address for the benefit of the reader.

1) First and foremost, I am not sure why the authors touch upon the issue of Planckian scattering rates at all, especially given the fact that the extracted prefactor (a) is quite small. Empirically, the observation of Planckian behavior is most interesting when the prefactor is an $O(1)$ number, from the point of view of various conjectured bounds. (There is no rigorously proven bound and we do not know what the prefactor should be even if such a bound exists.) The relevance of " a " and the emphasis on it in this paper was not clear to me from the point of view of the experimental discussion. There are other systems, e.g. monolayer graphene which exhibits T-linear resistivity above the Bloch Gruneisen temperature and exhibit "Planckian" scattering with similarly small coefficients. I am not implying that YRS exhibits T-linear resistivity due to el-ph scattering, but the Planckian aspect of the story seems like a distraction to me!

The authors write "...calling for further scrutiny of this concept in the limit of extreme correlation strength (we estimated effective masses above $1000m_0$ already at 0 T, which further increase towards the QCP), where transport is far from Drude like." I agree that Planckian scattering deserves further scrutiny, but the role of effective mass should ultimately drop out of the consideration of the scattering rate and other systems (e.g. Cuprates) which also do not have Drude like transport exhibit $a \sim O(1)$. Is there any optical conductivity data and measurement of optical scattering rates in YRS, which might shed complementary light on this issue?

2) More importantly, I am not entirely sure how the coefficient was extracted in the first place. Given the complicated multi-band nature and various associated subtleties, a clear discussion of the precise protocol that was adopted to extract the coefficient and the various caveats should be presented explicitly. In particular, for such multi-band systems, it is quite likely that only the (relatively) lighter excitations (as opposed to all the excitations) are responsible for charge transport. I am assuming this has not been factored into the analysis? A comment for the readers would be useful.

Reviewer #2:

Remarks to the Author:

In this study, the authors report their electrical resistivity and magneto resistivity measurements on 171/173YbRh₂Si₂ and 174YbRh₂Si₂ heavy-fermion superconductors in the temperature regime down to micro-Kelvin. They discovered that the superconducting state of 171/173YbRh₂Si₂ can be suppressed at a 60mT where the ground state shows an exotic strange metal state. Moreover, they found different superconducting behaviors under the magnetic fields for the two superconductors investigated, one is with a nuclear moment in Yb and the other is without a nuclear moment in Yb. Since the experimental temperature can go into a quite low regime, the results are interesting and important for understanding some key issues related to the connection between superconductivity and linear resistivity. However, this manuscript is failed to appropriately present the obtained experimental results in a logical and well-organized way. Therefore, I cannot recommend publication of the paper in NC in its present form. The flowing questions should be considered for the revised version:

1. The main interest to study the T-linear resistivity is aroused by that many unconventional superconductors show the maximum T_c when their normal state resistivity is T-linear. On the contrary, this study finds the T-linear resistivity in the ground state at QCP from superconducting to non-superconducting phases transition. The different T-linear resistivity behaviors found in this study with that of other unconventional superconductors should be one of the main contributions of this work, however it is not well discussed in the manuscript. In addition, is the normal state resistivity of the two superconductors studied also displays T-linear behavior? Authors should provide these data.
2. The analyses on the isotope effect on the QCP and T-linear resistivity are not presented clearly, which is important to understand the different superconducting behaviors observed under magnetic fields.
3. The reproducibility of the experimental results should be given in the revised version.

Reviewer #3:

Remarks to the Author:

This manuscript reports the results of ultra-low temperature measurements of the electrical resistivity of two high-quality single crystal samples of $\text{Yb}_2\text{Rh}_2\text{Si}_2$; one in which the Yb is of natural isotopic abundance and one in which the isotope is restricted to ^{174}Yb . The nominal distinction between the two samples is the presence of Yb nuclear moments in the former and absence in the latter. The principal observation is that of superconductivity in both samples. The superconducting state is further explored using magneto-thermal measurements of the resistivity to generate a temperature-magnetic field phase diagram for each material. This material is notable for the existence of a magnetic quantum critical point (QCP), when antiferromagnetic ordering is suppressed at zero temperature using a magnetic field. The key to this study is in tying the superconductivity, the QCP and the observed strange-metal, linear-in-temperature resistivity of the normal state together. Collectively, these properties are observed in a number of systems, transcending the details of their structure, and understanding exactly how they are related would be a significant step forward in this field.

The experimental work is exquisite, with the ultra-low temperature resistance measurements a technical tour-de-force. However, I cannot recommend publication in Nature Physics because in my opinion, there is insufficiently strong evidence to support the claims of the authors. I justify my opinion in the following paragraphs.

The observation of superconductivity alone is insufficient to warrant publication in this journal as it has already been reported in this material in the journal, Science, in 2016 (Ref 33). Admittedly, the observation of superconductivity in the isotopically pure material, the measurement of the superconducting phase diagram and the extension of the linear-in-temperature resistivity (strange metal) to lower temperatures are all new results. However, in my mind, they are incremental in nature according to the typical flow of the field after the initial discovery.

Beyond this, the authors claim to link the superconductivity to the well-established quantum critical point and establish the superconductivity as evolving from the strange metal normal state. Undoubtedly, this would be an important step for the field. However, I do not believe that the data supports this claim. Looking at the phase diagram in fig 4, the overwhelming message is that superconductivity is strongly suppressed at the QCP, if not killed entirely. The only evidence to support superconductivity at the QCP is the slight suppression (at the few percent level) of the resistance in either sample (see Figs 2a and 2b). To put this in context, the canonical unconventional superconductivity-QCP systems (e.g. CeIn_3 and High- T_c) have superconductivity nearly maximized (in terms of T_c) at the QCP. In this case it appears to me more likely that the QCP is antagonistic to superconductivity rather than the source of it. Furthermore, the lack of convincing evidence for

superconductivity at the QCP does not support the claim that superconductivity evolves from the strange metal state that exhibits linear-in-temperature resistivity. In fact superconductivity is strongest, in that it has the highest T_c , where the resistivity exhibits T^2 behaviour associated with a Fermi liquid state.

While the idea that the superconducting state is unconventional in nature seems natural, I am looking for experimental evidence to support this statement. The authors use the fact that the rate of change of B_{c2} with temperature ($-dB_{c2}/dT$) is large as one of the main experimental results to support this idea. However, I am not aware that this is a widely used criteria for unconventionality. The problem I see is that the temperature dependence of the resistivity in the mixed state of a superconductor involves contributions from flux flow when the motion of superconducting vortices generate voltages that mimic those of normal state resistance. Without understanding the nature of this, which is typically very sample dependent due to flux pinning relating to disorder, it is difficult to draw conclusive statements about the underlying superconducting state. This kind of physics may also be relevant to the differences in the phase diagrams derived from the two types of samples, and in reconciling phase diagrams deduced from resistivity measurements in this work and those from magnetic measurements on purportedly identical samples reported earlier (Ref 33). Finally, there is quite a bit of discussion towards the end of the paper concerning triplet superconductivity. Since there is absolutely no experimental evidence in this work to support any claim of this nature, this is highly speculative and I find the weight given to the discussion largely inappropriate.

To summarize, I do not support publication of the current manuscript in Nature Communications because I do not find that the data presented supports the broad conclusions that the authors need to make to bring the work to necessary level of impact for this journal. Moreover, the data itself is insufficient to warrant publication because it is supplemental to the original discovery of superconductivity in this material, which was reported some years earlier.

Point-by-point reply

Reviewer 1

I have read the paper by Nguyen et al. with a great deal of interest. The interplay of Planckian transport in strange metals and associated “high-temperature” superconductivity has gotten a great deal of attention in recent years, especially after Ref. 40 of the present manuscript pointed out the surprising universality associated with Drude scattering rates across microscopically distinct materials.

The field-tuned transport data presented in this manuscript is absolutely striking. The data presented in Figs. 2 and 3, especially the near-perfect T-linear behavior of the resistivity down to such low temperatures is remarkable and certainly deserves to be published. (I am not up to date on the latest experimental literature regarding YRS. I am assuming that the data presented here down to such low temperatures doesn’t exist elsewhere in the literature). However, I have a few questions that the authors might want to address for the benefit of the reader.

We thank the Reviewer for acknowledging the “absolutely striking” nature of our experiments at ultralow temperatures (accessible only via nuclear demagnetization cooling). These are indeed not only the first electrical resistivity measurements in this temperature range for YbRh_2Si_2 , but for any metal.

- 1.1** *First and foremost, I am not sure why the authors touch upon the issue of Planckian scattering rates at all, especially given the fact that the extracted prefactor (a) is quite small. Empirically, the observation of Planckian behavior is most interesting when the prefactor is an $O(1)$ number, from the point of view of various conjectured bounds. (There is no rigorously proven bound and we do not know what the prefactor should be even if such a bound exists.) The relevance of “ a ” and the emphasis on it in this paper was not clear to me from the point of view of the experimental discussion. There are other systems, e.g. monolayer graphene which exhibits T-linear resistivity above the Bloch Gruneisen temperature and exhibit “Planckian” scattering with similarly small coefficients. I am not implying that YRS exhibits T-linear resistivity due to el-ph scattering, but the Planckian aspect of the story seems like a distraction to me!*

The authors write “...calling for further scrutiny of this concept in the limit of extreme correlation strength (we estimated effective masses above 1000 m_0 already at 0 T, which further increase towards the QCP), where transport is far from Drude like.” I agree that Planckian scattering deserves further scrutiny, but the role of effective mass should ultimately drop out of the consideration of the scattering rate and other systems (e.g. Cuprates) which also do not have Drude like transport exhibit a $\sim O(1)$. Is there any optical conductivity data and measurement of optical scattering rates in YRS, which might shed complementary light on this issue?

The reason why we touch upon the issue of “Planckian dissipation” is that it has been attracting quite some attention in the community (as also the Reviewer states) and thus we felt that it would be appropriate to inform the reader about how the present “extreme strange metal” YbRh_2Si_2 relates to this concept. In a material with Planckian dissipation, a linear-in-temperature electrical resistivity arises when the scattering rate $1/\tau$ reaches the Planckian limit, $k_B T/\hbar$. Conversely, if linear-in-temperature electrical resistivity is seen even though $1/\tau$ is much smaller than the Planckian limit, as observed here, it might be caused by different physics. We hope the Reviewer agrees that this interesting finding deserves some attention.

We reiterate that electron-phonon scattering can be ruled out as the origin of the linear-in-temperature resistivity in YbRh_2Si_2 because (i) it is seen at very low temperatures where phonons are not expected to play an important role, (ii) the non- f reference material LuRh_2Si_2 is a normal metal, and (iii) Fermi liquid behavior is recovered when YbRh_2Si_2 is tuned away from the quantum critical point by magnetic field (see 2nd paragraph, page 3 of the manuscript).

We thank the Reviewer for his/her question regarding the optical scattering rate. Indeed, recent terahertz experiments revealed a linear-in-frequency inelastic optical scattering rate [see Fig. S4 of the supplementary materials of Science **367** (2020) 285, Ref. 12 of the manuscript]. By taking the ratio of the slopes $A' = \Delta\rho/\Delta T$ of the linear-in-temperature (dc) electrical resistivity and $A'' = \Delta[1/\text{Re}(\sigma_{\text{in}})]/\Delta\nu$ of the linear-in-frequency “optical resistivity”, all material-specific parameters drop out and the prefactor a of $k_{\text{B}}T/\hbar$ can be directly determined. The value of 0.0062 obtained in this way is in strikingly good agreement with our previous estimates (0.011 and 0.0065 for YbRh_2Si_2 and $^{174}\text{YbRh}_2\text{Si}_2$, respectively; Table II of the manuscript).

Changes: We have now included this new estimate of the prefactor (which we now call α instead of a , for consistency with most of the literature) in the Supplementary Information, Sect. S2 (see also 1.2 below). In addition, we have revised the statement about Planckian dissipation in YbRh_2Si_2 in the main part of the manuscript (top paragraph, page 8).

- 1.2** *More importantly, I am not entirely sure how the coefficient was extracted in the first place. Given the complicated multi-band nature and various associated subtleties, a clear discussion of the precise protocol that was adopted to extract the coefficient and the various caveats should be presented explicitly. In particular, for such multi-band systems, it is quite likely that only the (relatively) lighter excitations (as opposed to all the excitations) are responsible for charge transport. I am assuming this has not been factored into the analysis? A comment for the readers would be useful.*

Our procedure to determine the coefficient a (now α) was described in the caption of Table II. In fact, we were careful to avoid mistakes resulting from the multi-band nature of YbRh_2Si_2 by determining the average Fermi wavevector not from the Hall effect but from the Ginzburg-Landau coherence length, as already done in pioneering work on heavy fermion superconductors [Rauchschwalbe et al., Phys. Rev. Lett. **49**, 1448 (1982)]. We have now expanded this description further (see changes).

We note that we do not concur with the statement that the “(relatively) lighter excitations” are responsible for charge transport. The Kondo effect in general and Kondo destruction quantum criticality in particular are local phenomena, thus entailing all original conduction electrons near the Fermi level. If there were “light” carriers in heavy fermion metals that did not experience a Kondo effect, then these would dominate the electrical resistivity. This is clearly not the case, as beautifully demonstrated by Kadowaki-Woods scaling plots [revealing a universal A/γ^2 value, where $A \sim (m^*)^2$ is the linear-in- T^2 (Fermi liquid) resistivity coefficient, $\gamma \sim m^*$ the specific heat Sommerfeld coefficient, and m^* the renormalized effective mass; see, e.g., Tsujii et al., Phys. Rev. Lett. 94 (2005) 057201].

The good agreement of our (previous) estimate of the coefficient α with the independent new one from the optical conductivity data (discussed in 1.1 above) further validates this understanding.

Changes: We now describe the estimate of α from the experiments of the present work (together with the independent new estimate derived from optics, see point 1.1 above) in some detail in the Supplementary Information, Sect. S2. To avoid redundancy, we have adjusted the captions of Tables I and II accordingly.

We hope that the Reviewer finds our replies appealing and will support publication of the revised version, to push further the frontier of this important field.

Reviewer 2

In this study, the authors report their electrical resistivity and magneto resistivity measurements on $171/173\text{YbRh}_2\text{Si}_2$ and $174\text{YbRh}_2\text{Si}_2$ heavy-fermion superconductors in the temperature regime down to micro-Kelvin. They discovered that the superconducting state of $171/173\text{YbRh}_2\text{Si}_2$ can be suppressed at a 60mT where the ground state shows an exotic strange metal state. Moreover, they found different superconducting behaviors under the magnetic fields for the two superconductors investigated, one is with a nuclear moment in Yb and the other is without a nuclear moment in Yb. Since the experimental temperature can go into a quite low regime, the results are interesting and important for understanding some key issues related to the connection between superconductivity and linear resistivity. However, this manuscript is failed to appropriately present the obtained experimental results in a logical and well-organized way. Therefore, I cannot recommend publication of the paper in NC in its present form. The following questions should be considered for the revised version:

We thank also this Reviewer for acknowledging the interest and importance of our results, and are glad to follow his/her suggestions on how to further improve the presentation to make the work more accessible.

2.1 a. The main interest to study the T -linear resistivity is aroused by that many unconventional superconductors show the maximum T_c when their normal state resistivity is T -linear. On the contrary, this study finds the T -linear resistivity in the ground state at QCP from superconducting to non-superconducting phases transition.

We thank the Reviewer for this important point. Indeed, what comes to one's mind when thinking about unconventional superconductors is a "dome" of superconductivity centered around a quantum critical point (QCP), from which non-Fermi liquid behavior emerges. This has been observed in a number of materials tuned by (external or chemical) *pressure* (as well as by doping) across a QCP, as sketched in Fig. R1 left. Magnetic field then smoothly suppresses superconductivity (Fig. R1 right). The reason is that a magnetic field is, in general, hostile to superconductivity (due to both the orbital and Pauli-limiting effects).

FIG. R1: **Canonical situation of heavy fermion superconductors.** (left) A superconducting dome covers the pressure-tuned quantum critical point at P_c . (right) At P_c , superconductivity is smoothly suppressed by a magnetic field. From Flouquet et al., J. Phys.: Conf. Ser. **273** (2011) 012001.

YbRh_2Si_2 , by contrast, exhibits a *magnetic-field-induced* QCP. In this case, there are two competing effects: the suppression of T_c by the above "hostile" magnetic field effect and the enhancement of T_c by approaching the QCP. We think that our data support that this is what happens here, by

- the highly unusual shape of the $T_c(B)$ data (Fig. 4 of the manuscript, full and open data points) compared with that of Fig. R1 right;

- the large magnitude of the critical field when compared with both the upper critical field slopes (red lines) and the mean-field expectations (dashed blue and green lines);
- the fact that, at the quantum critical field of 60 mT (where linear-in- T resistivity holds down to the lowest temperatures), T_c of YbRh_2Si_2 is still almost 1 mK, clearly finite on our scales;
- the associated highly unusual $\rho(B)$ isotherms (see Fig. 3a,b of the manuscript), with a “double-increase” structure, which strongly suggests the presence of two superconducting phases for YbRh_2Si_2 and directly demonstrates it for $^{174}\text{YbRh}_2\text{Si}_2$.

In Fig. R2 we provide cartoons meant to help visualize the “hostile” effect a magnetic field can have on superconductivity mediated by fluctuations from a magnetic-field induced QCP.

FIG. R2: Cartoon of magnetic-field effect on a superconducting dome around a quantum critical point (QCP). **(left)** The fat blue curve is a hypothetical $T_c(B)$ curve where any “hostile” effect of the magnetic field on superconductivity (from Pauli or orbital limiting) is imagined to be absent. For the successive curves (from blue to green), an increasingly strong “hostile” field effect is considered (using a simple mean-field-type suppression of T_c by the magnetic field as shown in the right panel of Fig. R1). **(center)** Same as **left** but for a superconducting phase with a dome that does not extend to the zero of the tuning parameter axis (fat red curve). Again, an increasingly strong “hostile” field effect is added for the successive curves (from red to brown). **(right)** Lowest curve from the **left** panel (green) and middle curve of the **center** panel (purple), scaled in absolute values. A material with two QCP-derived superconducting phases, one with stronger pairing but larger field sensitivity (as in the **left** panel, e.g., spin singlet) and one with weaker pairing but also weaker field sensitivity (as in the **center** panel, e.g., spin triplet), would display such a phase diagram.

Changes: We suspect that the phase diagram, with data of both YbRh_2Si_2 and $^{174}\text{YbRh}_2\text{Si}_2$ in a single graph (Fig. 4 of previous manuscript), contained too much information to be well readable. Thus, we have now split the information into two separate panels. We hope that this will make our key results more accessible. In addition, we have expanded the description of Fig. 4 (bottom paragraph on page 4) and now explicitly remark on the “hostile” effect of magnetic fields. Finally, we have added Fig. R2 as new Fig. S3 to the Supplementary Information and refer to it in the main text. That a superconducting dome centered around a magnetic-field-induced QCP is, in general, *not* expected was already mentioned in the conclusion (3rd paragraph, page 7).

b. The different T -linear resistivity behaviors found in this study with that of other unconventional superconductors should be one of the main contributions of this work, however it is not well discussed in the manuscript.

We assume that the Reviewer is referring here to our observation of linear-in-temperature resistivity that, unlike in other materials, does not appear to be limited by Planckian dissi-

pation. We are sorry for having been too brief on this point. We have now considerably expanded the description. In addition, motivated by Reviewer 1's question on optical conductivity data (see point 1.1 above), we have also added a second, independent way of determining the proportionality coefficient in the Planckian scattering rate relation and find very good agreement with our original estimate. This further supports our original finding.

Changes: We have added Sect. 2 in the Supplementary Information, describing both the original estimate of the Planckian scattering rate and the comparison with the optical conductivity (please see point 1.1 above for further details).

c. In addition, is the normal state resistivity of the two superconductors studied also displays T -linear behavior? Authors should provide these data.

Yes, the two superconductors investigated here (results displayed in Figs. 3,4) are the very same samples showing linear-in- T resistivity at higher temperatures, in their normal state (results of Fig. 2). In fact, Fig. 1a,b gives an overview: Linear behavior at high temperatures (red shaded line), T^2 behavior below the Néel temperature (green shaded line), and the drop of resistivity signaling the onset of superconductivity (cut off y-axis to better resolve the high- T results) at the lowest temperatures.

Changes: We have slightly revised the description of the figures (on page 4 and 5 of the manuscript) to better guide the reader through our findings.

2.2 *The analyses on the isotope effect on the QCP and T -linear resistivity are not presented clearly, which is important to understand the different superconducting behaviors observed under magnetic fields.*

The two different samples are labeled “ YbRh_2Si_2 ” (natural abundance Yb isotope mixture) and “ $^{174}\text{YbRh}_2\text{Si}_2$ ” (isotope-pure Yb 174). All measurements were done on both samples and are shown in Fig. 1: a,c for YbRh_2Si_2 , b,d for $^{174}\text{YbRh}_2\text{Si}_2$; Fig. 2: a for YbRh_2Si_2 , b for $^{174}\text{YbRh}_2\text{Si}_2$, c for both; Fig. 3: a,c for YbRh_2Si_2 , b,d for $^{174}\text{YbRh}_2\text{Si}_2$; revised Fig. 4: a for YbRh_2Si_2 , b for $^{174}\text{YbRh}_2\text{Si}_2$.

Changes: See changes in 2.1c. In addition we hope that the splitting of Fig. 4 into two panels, (a) for YbRh_2Si_2 and (b) for $^{174}\text{YbRh}_2\text{Si}_2$, has improved the clarity.

2.3 *The reproducibility of the experimental results should be given in the revised version.*

Indeed, reproducibility of experiments is an important point, and may be nontrivial at ultralow temperatures. This is why we performed all measurements with great care. Many of the data were taken multiple times and were confirmed to be fully reproducible. A direct demonstration thereof is seen in Fig. 4. The full data points (of all three colors) represent resistive transitions as determined by isothermal field sweeps, the open symbols (again of all three colors) represent isofield temperature sweep. The fact that there is excellent agreement between these data is very strong evidence for the high reproducibility of our results. This was already noted in the Methods, Sect. E, lines 377-378.

As to sample reproducibility, we used single crystals from batches that have been studied in depth previously, and were shown to be highly reproducible. This was already stated in the previous manuscript (now: page 3, lines 70, 75-77; caption of Table I).

Changes: We have now added a corresponding statement also to the caption of Fig. 4.

We hope that the Reviewer is satisfied by our replies, finds that our revisions have made the work more accessible, and can now recommend it for publication.

Reviewer 3

This manuscript reports the results of ultra-low temperature measurements of the electrical resistivity of two high-quality single crystal samples of $\text{Yb}_2\text{Rh}_2\text{Si}_2$; one in which the Yb is of natural isotopic abundance and one in which the isotope is restricted to ^{174}Yb . The nominal distinction between the two samples is the presence of Yb nuclear moments in the former and absence in the latter. The principal observation is that of superconductivity in both samples. The superconducting state is further explored using magneto-thermal measurements of the resistivity to generate a temperature-magnetic field phase diagram for each material. This material is notable for the existence of a magnetic quantum critical point (QCP), when antiferromagnetic ordering is suppressed at zero temperature using a magnetic field. The key to this study is in tying the superconductivity, the QCP and the observed strange-metal, linear-in-temperature resistivity of the normal state together.

Collectively, these properties are observed in a number of systems, transcending the details of their structure, and understanding exactly how they are related would be a significant step forward in this field.

The experimental work is exquisite, with the ultra-low temperature resistance measurements a technical tour-de-force.

We thank the Reviewer for acknowledging the importance of the topic studied and the “exquisite” nature of our ultralow-temperature experiments.

However, I cannot recommend publication in Nature Physics because in my opinion, there is insufficiently strong evidence to support the claims of the authors. I justify my opinion in the following paragraphs.

3.1 *The observation of superconductivity alone is insufficient to warrant publication in this journal as it has already been reported in this material in the journal, Science, in 2016 (Ref 33). Admittedly, the observation of superconductivity in the isotopically pure material, the measurement of the superconducting phase diagram and the extension of the linear-in-temperature resistivity (strange metal) to lower temperatures are all new results. However, in my mind, they are incremental in nature according to the typical flow of the field after the initial discovery.*

We thank the Reviewer for acknowledging our experimental achievements, namely:

- the discovery of superconductivity in isotope pure YbRh_2Si_2 (referred to as $^{174}\text{YbRh}_2\text{Si}_2$);
- the measurement of the superconducting T - B phase diagrams (below 15 mK and up to 70 mT) of YbRh_2Si_2 and $^{174}\text{YbRh}_2\text{Si}_2$;
- the expansion of the linear-in- T range of the electrical resistivity of both YbRh_2Si_2 and $^{174}\text{YbRh}_2\text{Si}_2$ to 3.5 orders of magnitude in temperature.

This list could be further expanded to include:

- the first ever electrical resistivity measurement of YbRh_2Si_2 below about 15 mK and, as a matter of fact, of any metal at ultralow temperatures, opening a new chapter in ultralow temperature physics;
- the discovery of an extreme strange metal that is not governed by the Planckian limit.

This allowed us to draw conclusions that could not have been anticipated from the Science 2016 work (Ref. 33), namely that:

- superconductivity in YbRh_2Si_2 extends to surprisingly large fields, even a factor of 3 larger than the already large estimate of the critical field (B'_{c2} , see Table II) from the upper critical field slope ($-dB_{c2}/dT|_{T_c}$, see also Table II). This is a strong indication for a mechanism boosting superconductivity at *finite* fields (i.e., the QCP) and for the unconventional nature of the superconductivity.
- the boundary of the superconducting phase of YbRh_2Si_2 is highly “non-mean-field-like” (compare with Fig. R1 right of point 2.1 above), suggesting a two-phase nature, with the high-field phase having all necessary ingredients for spin-triplet superconductivity. This will trigger further experiments to test this tentative assignment.
- in the absence of Yb nuclear moments the superconductivity is severely weakened, indicating that nuclear moments *do* play a role in *strengthening* superconductivity, but are not needed for *creating* it. This puts concise constraints on any microscopic theory for the observed superconductivity.
- the phase diagram of $^{174}\text{YbRh}_2\text{Si}_2$ clearly falls in two different phases, supporting the two-phase assignment of the superconducting phase region in YbRh_2Si_2 .
- quantum critical fluctuations, with beyond-order parameter nature as demonstrated previously, are responsible for both the extreme strange metal behavior (over 3.5 orders of magnitude in temperature) and for (at least a component of) the superconductivity of YbRh_2Si_2 , thereby pointing to a microscopic mechanism of strange metal superconductivity. We hope that our revisions (see also points 3.2-4 below) have in particular helped to underpin this last key point.

In our opinion this is a set of important (non-incremental) discoveries, and we trust that they will trigger further important work, both theoretical and experimental.

3.2 *Beyond this, the authors claim to link the superconductivity to the well-established quantum critical point and establish the superconductivity as evolving from the strange metal normal state. Undoubtedly, this would be an important step for the field. However, I do not believe that the data supports this claim. Looking at the phase diagram in fig 4, the overwhelming message is that superconductivity is strongly suppressed at the QCP, if not killed entirely. The only evidence to support superconductivity at the QCP is the slight suppression (at the few present level) of the resistance in either sample (see Figs 2a and 2b). To put this in context, the canonical unconventional superconductivity-QCP systems (e.g. CeIn_3 and High- T_c) have superconductivity nearly maximized (in terms of T_c) at the QCP. In this case it appears to me more likely that the QCP is antagonistic to superconductivity rather than the source of it. Furthermore, the lack of convincing evidence for superconductivity at the QCP does not support the claim that superconductivity evolves from the strange metal state that exhibits linear-in-temperature resistivity. In fact superconductivity is strongest, in that it has the highest T_c , where the resistivity exhibits T^2 behaviour associated with a Fermi liquid state.*

In this reasoning the Reviewer overlooks the fact that we are here using magnetic field to tune the material through its QCP, and not (chemical or external) pressure or doping as in CeIn_3 or the high- T_c 's. Because magnetic field generally suppresses superconductivity (due to the Pauli-limiting and/or orbital effect), there are now two competing trends: the suppression of T_c by magnetic field and the enhancement of T_c by approaching the QCP. We think that the shapes of the $T_c(B)$ curves shown in Fig. 4 (full and open data points) support that this is what happens here. Further details, and in particular a visualization of this effect

with simple cartoons (Fig. R2), are given in point 2.1 above, to which we kindly refer the Reviewer.

We also note that the Reviewer’s statement about the slight (“at the few percent level”) suppression of the resistivity at the QCP of YbRh_2Si_2 is incorrect, and are sorry for the unclear description that might have caused this confusion. Figure 2 (entitled “Strange metal behavior of ...”) highlights the linear-in- T behavior and thus the resistivity axis was set to just show the onset of superconductivity. The full resistivity curves are shown in Fig. 1. For YbRh_2Si_2 , at the quantum critical field of 60 mT, the resistivity has dropped to 50% at somewhat below 1 mK (Fig. 1c, green curve). Thus, T_c of YbRh_2Si_2 is clearly finite at the QCP. Moreover, the pale shadings in Fig. 4 have real meanings: they represent 90% resistance lines (thus a substantial drop of 10%). For YbRh_2Si_2 (light blue shading) there is a clear local maximum in this curve near the quantum critical field.

Changes: We have now separated the phase diagrams of YbRh_2Si_2 and $^{174}\text{YbRh}_2\text{Si}_2$ (previous Fig. 4) into two panels (new Fig. 4a,b), hoping that this increases the readability. The border of the dark blue shading—a guide-to-the-eyes for $T_c(B)$ in the original manuscript—now has more concrete meaning near the QCP: It corresponds to a linear fit to the measured high-field data points, and further underpins superconductivity extending beyond the QCP. In addition, we have expanded the description of Fig. 4 in the main text (bottom paragraph on page 4) and now explicitly remark on the “hostile” effect of magnetic fields. Finally, we have added Fig. R2 as new Fig. S3 to the Supplementary Information and refer to it in the main text. That a superconducting dome centered around a magnetic-field-induced QCP is, in general, *not* expected was already mentioned in the conclusion (3rd paragraph, page 7). Finally, to avoid the above misreading of our data (“at the few percent level”), we have improved the description of superconductivity *at* the QCP (first paragraph, page 4).

3.3 *While the idea that the superconducting state is unconventional in nature seems natural, I am looking for experimental evidence to support this statement. The authors use the fact that the rate of change of B_{c2} with temperature ($-dB_{c2}/dT$) is large as one of the main experimental results to support this idea. However, I am not aware that this is a widely used criteria for unconventionality. The problem I see is that the temperature dependence of the resistivity in the mixed state of a superconductor involves contributions from flux flow when the motion of superconducting vortices generate voltages that mimic those of normal state resistance. Without understanding the nature of this, which is typically very sample dependent due to flux pinning relating to disorder, it is difficult to draw conclusive statements about the underlying superconducting state. This kind of physics may also be relevant to the differences in the phase diagrams derived from the two types of samples, and in reconciling phase diagrams deduced from resistivity measurements in this work and those from magnetic measurements on purportedly identical samples reported earlier (Ref 33).*

Already the “historic” work on the first heavy fermion superconductor CeCu_2Si_2 reports an unusually large value of the upper critical field slope $-dB_{c2}/dT|_{T_c}$ and associates it with the high density of Cooper pairs resulting from the heavy fermion quasiparticles of the heavy fermion normal state [Rauchschwalbe et al., Phys. Rev. Lett. **49**, 1448 (1982)]. The direct correlation between $-dB_{c2}/dT|_{T_c}$ and the Sommerfeld coefficient of the specific heat (which is proportional to the electronic density of states at the Fermi level) was discovered even earlier in the chevrel phases [Ø. Fischer, Appl. Phys. **16** (1978) 1]. This understanding is still state-of-the-art.

In addition, it is the boosting of superconductivity towards the QCP—against the “hostile” effect of magnetic fields on superconductivity (see point 3.2 and 2.1 with Fig. R2 above)—that provides strong evidence for quantum critical fluctuations governing at least part of the superconductivity in YbRh_2Si_2 .

Flux-flow resistivity can be identified via a current-density-dependent electrical resistivity in the (partially) superconducting state. We have studied this effect during our initial experiments, though not having possible flux-flow effects in mind but the determination of the highest possible measurement current that does not lead to sample overheating. In Fig. R3 we show such experiments on both YbRh_2Si_2 and $^{174}\text{YbRh}_2\text{Si}_2$. The only difference between $R(T)$ [or $R(B)$] curves measured with different currents is that data measured with lower currents are noisier (the displayed data were taken with same the statistics), and that the application of too large currents leads to overheating (slightly higher resistance at low temperatures). There is, however, no indication for a flux-flow resistance.

FIG. R3: **Test measurements of current dependence of resistance.** (left) The temperature-dependent resistance data of YbRh_2Si_2 were taken at 17 mT, with excitation currents between 6 nA and 200 nA, in order to find the best current settings. (right) The field-dependent resistance data of $^{174}\text{YbRh}_2\text{Si}_2$ were taken at 1.2 mK, with two different currents. No flux-flow resistance contributions can be identified. All data presented in the manuscript were taken with currents that do not overheat the sample (10 nA at the lowest temperatures), with much longer averaging time (and improved electronics/shielding) than these preliminary measurements.

With respect to the Reviewer’s last point we note that the agreement between the present work and results from ref. 33 is quite satisfying.

Changes: We have added comments on flux flow (page 4, 2nd paragraph and Methods, Sect. E, last sentence) and improved the paragraph on the comparison with ref. 33 (page 6, 1st paragraph). In addition, we have included a new figure to the Supplementary Information (Fig. S2) that sketches two possible scenarios that are compatible with the combined results of ref. 33 and the present work, and refer to it in the text (also page 6, 1st paragraph).

3.4 *Finally, there is quite a bit of discussion towards the end of the paper concerning triplet superconductivity. Since there is absolutely no experimental evidence in this work to support any claim of this nature, this is highly speculative and I find the weight given to the discussion largely inappropriate.*

We certainly agree with the Reviewer that our evidence for spin-triplet superconductivity in YbRh_2Si_2 and $^{174}\text{YbRh}_2\text{Si}_2$ is only tentative, and we think that our concluding statement

(“Naturally, the proposal of spin-triplet superconductivity should also be scrutinized by future experiments, including probes of anisotropies and NMR investigations, which are in principle feasible at ultralow temperatures.”, see page 7, end of 2nd paragraph) should have made this sufficiently clear. Let us explain why we still think that it is interesting and appropriate to discuss this possibility. Spin-triplet superconductivity is notoriously difficult to firmly pin down, as shown maybe most prominently by the case of Sr_2RuO_4 , which had been considered as key “candidate” spin-triplet superconductor for many years but was recently concluded not to be one [Pustogow et al., *Nature* **574** (2019) 72]. It is common practice to use the comparison of the measured critical field and the Pauli limiting upper critical field as first indication for spin-triplet superconductivity. The fact that the upper critical field determined for YbRh_2Si_2 (> 60 mT, Fig. 4a) exceeds by more than a factor of four the usual Pauli limit (15 mT, see Table II) is thus a first hint that spin-triplet superconductivity could be realized.

Here, we provide two additional pieces of evidence. Firstly, in $^{174}\text{YbRh}_2\text{Si}_2$, the phase diagram clearly displays two phases. This adds evidence that a change in pairing symmetry occurs, likely also in the electronically identical compound YbRh_2Si_2 (the cartoons we draw in Fig. S3 help to see how this situation can lead to the type of phase diagrams we observe). Secondly, as discussed in the manuscript (2nd paragraph on page 7), there is theoretical support for spin-triplet pairing becoming competitive in large enough magnetic fields (ref. 39 of the manuscript).

Just as side remark, we are committed to setting up NMR experiments at ultralow temperatures—another tour-de-force effort—to scrutinize this tentative assignment. Andrej Pustogow, who was just appointed at our institute, will join this effort.

To summarize, I do not support publication of the current manuscript in Nature Communications because I do not find that the data presented supports the broad conclusions that the authors need to make to bring the work to necessary level of impact for this journal. Moreover, the data itself is insufficient to warrant publication because it is supplemental to the original discovery of superconductivity in this material, which was reported some years earlier.

We hope that the Reviewer is satisfied with our replies, and finds that the changes to the manuscript have clarified both the level of achievement beyond the Science-2016 work and the connection between the presented data and conclusions drawn from them.

Reviewers' Comments:

Reviewer #1:

Remarks to the Author:

I have reviewed the detailed response of the authors to the questions raised by all the referees. I had already mentioned in my earlier report that this manuscript deserves to be published eventually. The other referees had been less enthusiastic and had raised valid questions, which the authors appear to have answered satisfactorily.

Hence, I would like to reiterate my earlier recommendation that the manuscript be accepted for publication in Nature communications.

Reviewer #2:

Remarks to the Author:

I have read the answers of the authors to my and other referee's comments. Overall I was satisfied with the clarifications made and the associated changes in the text. I therefore recommend publication in Nature Communications of this work.

Reviewer #3:

Remarks to the Author:

This is my report on the second revised version of the manuscript, "Deciphering high-temperature superconductivity with ultralow temperatures" and the accompanying, "point by point reply" document that is a response to the first round of referees comments.

I agree with the majority of the responses by the authors to the comments specifically to reviewer 3. Furthermore, incorporation of many of the reviewers comments means that the revised manuscript is improved in terms of clarity in comparison to initial submission.

However, I still find that the data presented does not unambiguously support the larger claims by the authors that would elevate the work beyond what is still an interesting report on low temperature transport properties of YbRh₂Si₂ in its natural and isotopically pure forms.

The key area of concern for me remains the establishment of the relationship between the linear in temperature resistivity and superconductivity in YbRh₂Si₂. In the fourth paragraph of the manuscript, the authors state that their report "establishes the connection between electron localization-delocalization-derived [48] strange metal behavior and unconventional superconductivity, discussed previously for several [49] other materials [21–24], to an unprecedented level of confidence,". This is a bold claim and requires considerable experimental scrutiny.

The data at hand to support this claim are provided in Figure 1. They are also the topic of point 3.2 in the rebuttal document, where the authors write, "The full resistivity curves are shown in Fig. 1. For YbRh₂Si₂, at the quantum critical field of 60mT, the resistivity has dropped to 50% at somewhat below 1mK (Fig. 1c, green curve). Thus, T_c of YbRh₂Si₂ is clearly finite at the QCP. Moreover, the pale shadings in Fig. 4 have real meanings: they represent 90% resistance lines (thus a substantial drop of 10%). For YbRh₂Si₂ (light blue shading) there is a clear local maximum in this curve near the quantum critical field."

An alternative description of the data would be the following. At the quantum critical field (60 mT), the resistivity of the natural sample of YbRh₂Si₂ at the lowest available measurement temperatures has fallen to 50% of the normal state value, indicating that a considerable amount of the sample remains

in the normal state. For the isotopically pure sample, the resistivity has dropped by only a few percent, indicating that the vast majority (>90%) of the sample is still in the normal state. Implicit here is the difficulty of knowing exactly how much of the sample is truly superconducting on the basis of resistivity measurements alone. Furthermore, the trajectory of the temperature dependences (at 60 mT) on the basis of the current data set does not lead to zero resistivity at any finite temperature.

The attribution of T_c to when the resistivity falls to 50% of its normal state value is a widely used definition, but this is usually in a situation where the full transition is available. To use it as a basis for defining a superconducting transition temperature in the absence of the full transition is less obvious since a definition that is arbitrarily lower, for example 40%, does not allow one to define a transition temperature in either sample at the quantum critical field.

The authors quite rightly state that the application of a magnetic field to YbRh_2Si_2 has two effects. On the one hand, it is the tuning parameter necessary for accessing the quantum critical point, while on the other it is conventionally antagonistic to superconductivity. This is simply an unavoidable (an in this case unfortunate) effect of nature, which makes superconductivity and quantum criticality difficult to study simultaneously in this material. So, while YbRh_2Si_2 may avoid the complexity of phase diagrams that hamper the deciphering of the relationship between superconductivity and the strange-metal normal state, it has its own unique issues concerning the dual effects of magnetic field.

For this reason, the authors are forced to infer that superconductivity evolves out of the strange metal state by bolstering the limitations of the data with models in which they compensate for the effect of the magnetic field on superconductivity. This is markedly different from the data itself providing an "unprecedented level of confidence" on this issue and the assuredness of the wide-reaching claims, which include the title of the manuscript itself.

In the end, it is not whether I believe that there is a relationship between the strange metal state and superconductivity in this material, but that the authors need to improve the clarity of what is unambiguous according to the data they present and what is an assumption. The bottom line here is that, on the basis of the data, the superconductivity in this material is strongest (in terms of highest T_c) in the Fermi-liquid (T_2) regime at zero magnetic field. It is either non-existent or only partially present at the lowest possible measurement temperatures at the quantum critical magnetic field. It is also an intriguing observation that in the natural sample of YbRh_2Si_2 the disappearance of superconductivity is at almost the exact field as the quantum critical point. Examining this issue might also require better knowledge of the precise value of the quantum critical field itself, about which the current manuscript is notably lacking. To balance the appraisal of the data, I think it is reasonable to expect the authors to address this perspective.

In summary, my recommendation is that current version of the manuscript is unsuitable for publication in its current form. The authors must clearly distinguish between what is factual according to experimental data and what conclusions are drawn based on reasonable assumptions. Of course, my expectation is that this will somewhat restrict the appeal of the manuscript, but given the extended list of experimental achievements provided in response 3.1, there is still a strong case for publication in Nature Communications (apologies for the reference to Nature Physics in my first report).

Point-by-point reply

Reviewer 1 – 2nd report

I have reviewed the detailed response of the authors to the questions raised by all the referees. I had already mentioned in my earlier report that this manuscript deserves to be published eventually. The other referees had been less enthusiastic and had raised valid questions, which the authors appear to have answered satisfactorily.

Hence, I would like to reiterate my earlier recommendation that the manuscript be accepted for publication in Nature communications.

We thank the Reviewer for going through all three reports and our replies and are glad to see that he/she is satisfied and recommends publication of our manuscript in its present form.

Reviewer 2 – 2nd report

I have read the answers of the authors to my and other referee's comments. Overall I was satisfied with the clarifications made and the associated changes in the text. I therefore recommend publication in Nature Communications of this work.

Many thanks also to the second Reviewer for evaluating our replies to all three Reviewers. Again we are glad that the Reviewer is satisfied with our replies and the associated changes to the manuscript and now recommends publication.

Reviewer 3 – 2nd report

This is my report on the second revised version of the manuscript, “Deciphering high-temperature superconductivity with ultralow temperatures” and the accompanying, “point by point reply” document that is a response to the first round of referees comments.

I agree with the majority of the responses by the authors to the comments specifically to reviewer 3. Furthermore, incorporation of many of the reviewers comments means that the revised manuscript is improved in terms of clarity in comparison to initial submission.

We are also grateful to Reviewer 3 for taking the time to evaluate the revised manuscript and our replies to his/her previous comments. We are glad to see that he/she is satisfied by most of our responses and concludes (see below) that “there is still a strong case for publication in Nature Communications”. Of course we gladly address his/her remaining concern. For clarity, we label the paragraphs of the present report (pasted below) by **a**, **b**, **c**... and refer to points of the Reviewer's previous report (appended at the end) by **3.1**, **3.2**, **3.2**....

a, *However, I still find that the data presented does not unambiguously support the larger claims by the authors that would elevate the work beyond what is still an interesting report on low temperature transport properties of YbRh₂Si₂ in its natural and isotopically pure forms.*

b, *The key area of concern for me remains the establishment of the relationship between the linear in temperature resistivity and superconductivity in YbRh₂Si₂. In the fourth paragraph of the manuscript, the authors state that their report “establishes the connection between electron localization-delocalization-derived [48] strange metal behavior and unconventional superconductivity, discussed previously for several [49] other materials [21-24], to an unprecedented level of confidence, ...”. This is a bold claim and requires considerable experimental scrutiny.*

c, *The data at hand to support this claim are provided in Figure 1. They are also the topic of point 3.2 in the rebuttal document, where the authors write, “The full resistivity curves are shown in Fig. 1. For YbRh₂Si₂, at the quantum critical field of 60mT, the resistivity has dropped to 50% at somewhat below 1mK (Fig. 1c, green curve). Thus, T_c of YbRh₂Si₂ is clearly finite at the QCP. Moreover, the pale shadings in Fig. 4 have real meanings: they represent 90% resistance lines (thus a substantial drop of 10%). For YbRh₂Si₂ (light blue shading) there is a clear local maximum in this curve near the quantum critical field.”*

d, *An alternative description of the data would be the following. At the quantum critical field (60 mT), the resistivity of the natural sample of YbRh₂Si₂ at the lowest available measurement temperatures has fallen to 50% of the normal state value, indicating that a considerable amount of the sample remains in the normal state. For the isotopically pure sample, the resistivity has dropped by only a few percent, indicating that the vast majority (> 90%) of the sample is still in the normal state. Implicit here is the difficulty of knowing exactly how much of the sample is truly superconducting on the basis of resistivity measurements alone. Furthermore, the trajectory of the temperature dependences (at 60 mT) on the basis of the current data set does not lead to zero resistivity at any finite temperature.*

e, *The attribution of T_c to when the resistivity falls to 50% of its normal state value is a widely used definition, but this is usually in a situation where the full transition is available. To use it as a basis for defining a superconducting transition temperature in the absence of the full transition is less obvious since a definition that is arbitrarily lower, for example 40%, does not allow one to define a transition temperature in either sample at the quantum critical field.*

f, *The authors quite rightly state that the application of a magnetic field to YbRh₂Si₂ has two effects. On the one hand, it is the tuning parameter necessary for accessing the quantum critical point, while on the other it is conventionally antagonistic to superconductivity. This is simply an unavoidable (and in this case unfortunate) effect of nature, which makes superconductivity and quantum criticality difficult to study simultaneously in this material. So, while YbRh₂Si₂ may avoid the complexity of phase diagrams that hamper the deciphering of the relationship between superconductivity and the strange-metal normal state, it has its own unique issues concerning the dual effects of magnetic field.*

g, *For this reason, the authors are forced to infer that superconductivity evolves out of the strange metal state by bolstering the limitations of the data with models in which they compensate for the effect of the magnetic field on superconductivity. This is markedly different from the data itself providing an “unprecedented level of confidence” on this issue and the assuredness of the wide-reaching claims, which include the title of the manuscript itself.*

h, *In the end, it is not whether I believe that there is a relationship between the strange metal state and superconductivity in this material, but that the authors need to improve the clarity of what is unambiguous according to the data they present and what is an assumption. The bottom line here is that, on the basis of the data, the superconductivity in this material is strongest (in terms of highest T_c) in the Fermi-liquid (T₂) regime at zero magnetic field. It is either non-existent or only partially present at the lowest possible measurement temperatures at the quantum critical magnetic field. It is also an intriguing observation that in the natural sample of YbRh₂Si₂ the disappearance of superconductivity is at almost the exact field as the quantum critical point. Examining this issue might also require better knowledge of the precise value of the quantum critical field itself, about which the current manuscript is notably lacking. To balance the appraisal of the data, I think it is reasonable to expect the authors to address this perspective.*

i, *In summary, my recommendation is that current version of the manuscript is unsuitable for*

publication in its current form. The authors must clearly distinguish between what is factual according to experimental data and what conclusions are drawn based on reasonable assumptions. Of course, my expectation is that this will somewhat restricts the appeal of the manuscript, but given the extended list of experimental achievements provided in response 3.1, there is still a strong case for publication in Nature Communications (apologies for the reference to Nature Physics in my first report).

In the first two paragraphs (**a** and **b**) the Reviewer expresses his/her remaining concern, and details it further in the following paragraphs (**c** - **h**).

We thank him/her for now summarizing our temperature-dependent electrical resistivity data from Fig. 1 correctly (**c**, see **3.2** for the previous oversight of the relevant panels). Note that important additional data, not mentioned by the Reviewer but equally used to construct the phase diagrams of Fig. 4, are the isothermal magnetic-field curves in Fig. 3 a, b, as well as the color-coded phase diagrams in Fig. 3 c, d (both discussed further below).

As to the discussion in **d** and **e**, we are not sure what the Reviewer wants to implicate. To us, there cannot be any reasonable doubt about the presence of superconductivity in our data. Then, the precise definition of T_c boils down to being a rather technical issue. Any superconducting transitions has a finite width and, as the Reviewer acknowledges in **e**, to define T_c at the half-height is a widely used criterion. We also note that a visual extrapolation on linear scales to judge whether or not the resistivity will fall to zero at lower temperatures may not be reliable for the natural temperature scale is logarithmic.

For our conclusions, however, another point is more important than the precise T_c definition. As seen from Fig. 3 a, b, below a certain temperature, the resistivity vs field isotherms show a “two-step” structure (that is not due to flux flow, as discussed in **3.3**). The lower temperature part of it is boosted by the magnetic field, *against* the general trend of field suppression that the Reviewer now acknowledges (**f**, and **3.2** for the previous oversight of the field suppression effect). The boosting effect is also seen in the totally unbiased color-coded plots in Fig. 3 c, d. We see no other reasonable explanation for it than that the proximity to the quantum critical point is responsible for this superconductivity. This is, to us, strong evidence for the intimate connection of quantum criticality and at least this high-field superconducting phase of YbRh_2Si_2 .

Stepping back, one should acknowledge that YbRh_2Si_2 is a material that, already natively (in zero field), is situated in extreme vicinity to a quantum critical point. The Néel temperature is only 70 mK and the quantum critical field is only 60 mT. Quantum critical behavior that emerges from the QCP (at 60 mT and 0 K) expands quickly over wide field ranges as temperature is increased; already at 90 mK, the quantum critical fan [orange “tornado” of Fig. 1a of Custers et al., Nature 424 (2003) 524] extends to zero field. As such it seems likely that even the low-field superconducting phase is due to quantum critical fluctuations and not by coincidence occurring in such vicinity to a QCP.

We had tried to illustrate this situation in the last round by adding the cartoons of Fig. S3 to the Supplementary Information. We paste panel b of this figure below because it helps to further illustrate the Reviewer’s point. The fat red curve is a hypothetical $T_c(B)$ curve where any “hostile” effect of the magnetic field on superconductivity (from Pauli or orbital limiting) is imagined to be absent. For the successive curves (from red to brown), an increasingly strong “hostile” field effect is considered (using a simple mean-field-type suppression of T_c by the magnetic field B , which increases from zero along the tuning parameter axis). The purple curve is roughly what we observe in YbRh_2Si_2 . The physics, however, would still be the same if we had observed a further

FIG. N1: **Replotted from FIG. S3 b.** Cartoon of the (mean-field like pair-breaking) magnetic-field effect on a superconducting dome around a quantum critical point (QCP).

suppressed dome (brownish curves).

On the discussion in **f** we only partially agree. Whereas magnetic field as control parameter to reach a QCP has the disadvantage of superimposing a pair-breaking effect, it helps to (i) distinguish phases of different symmetry (singlet vs triplet pairing) via their different field sensitivity. Furthermore, *because* superconductivity is weakened, (ii) normal state non-Fermi liquid behavior can be accessed down to lower temperatures (establishing the record span of 3.5 orders of magnitude of linear-in- T range, Fig. 2 c). Finally, (iii) magnetic field is a continuous and clean control parameter that is available even at ultralow temperatures. As such, we consider the setting of YbRh_2Si_2 rather fortunate.

As to point **g** we hope that the above discussion clarifies that we are not “bolstering the limitations of the data with models” but instead are using a transparent and standard T_c definition as well as fully unbiased color-coded plots. (Fits with the simple two-phase model of the Supplementary Information were only done for $^{174}\text{YbRh}_2\text{Si}_2$; they only affect the region between the two phases.)

Finally, we address the Reviewer’s comment in **h** that a “better knowledge of the precise value of the quantum critical field itself” would be needed. We are sorry for not having explicated this in the caption of Fig. 4. The most precise determination of the quantum critical field of YbRh_2Si_2 for the direction we study here ($B \perp c$) comes from Refs. 28,29. For convenience, we replot the relevant panels below. In fact, we were conservative in assuming a critical field value of 60 mT. The right panel in Fig. N2 may suggest an even smaller value. We have now amended the caption of our Fig. 4 accordingly.

We also gladly took up his/her recommendation (in **i**) to screen the manuscript for cautious wording and have amended the formulations at several places. These changes, as well as the amendment of the caption of Fig. 4 discussed above, are printed in red in an extra copy of the revised manuscript.

We hope that Reviewer 3 is satisfied with our explanations and the changes to the manuscript, and can now recommend publication of the manuscript.

FIG. N2: **Higher-temperature phase diagrams of YbRh_2Si_2 from the literature.** Both locate the QCP at 0.06 mT at maximum. Left panel from Ref. 28, right panel from Ref. 29. These single crystals, as well as ours, were all grown by Cornelius Krellner by the same technique (Ref. 32).

Reviewer 3

This manuscript reports the results of ultra-low temperature measurements of the electrical resistivity of two high-quality single crystal samples of $\text{Yb}_2\text{Rh}_2\text{Si}_2$; one in which the Yb is of natural isotopic abundance and one in which the isotope is restricted to ^{174}Yb . The nominal distinction between the two samples is the presence of Yb nuclear moments in the former and absence in the latter. The principal observation is that of superconductivity in both samples. The superconducting state is further explored using magneto-thermal measurements of the resistivity to generate a temperature-magnetic field phase diagram for each material. This material is notable for the existence of a magnetic quantum critical point (QCP), when antiferromagnetic ordering is suppressed at zero temperature using a magnetic field. The key to this study is in tying the superconductivity, the QCP and the observed strange-metal, linear-in-temperature resistivity of the normal state together.

Collectively, these properties are observed in a number of systems, transcending the details of their structure, and understanding exactly how they are related would be a significant step forward in this field.

The experimental work is exquisite, with the ultra-low temperature resistance measurements a technical tour-de-force.

We thank the Reviewer for acknowledging the importance of the topic studied and the “exquisite” nature of our ultralow-temperature experiments.

However, I cannot recommend publication in Nature Physics because in my opinion, there is insufficiently strong evidence to support the claims of the authors. I justify my opinion in the following paragraphs.

3.1 *The observation of superconductivity alone is insufficient to warrant publication in this journal as it has already been reported in this material in the journal, Science, in 2016 (Ref 33). Admittedly, the observation of superconductivity in the isotopically pure material, the measurement of the superconducting phase diagram and the extension of the linear-in-temperature resistivity (strange metal) to lower temperatures are all new results. However, in my mind, they are incremental in nature according to the typical flow of the field after the initial discovery.*

We thank the Reviewer for acknowledging our experimental achievements, namely:

- the discovery of superconductivity in isotope pure YbRh_2Si_2 (referred to as $^{174}\text{YbRh}_2\text{Si}_2$);
- the measurement of the superconducting T – B phase diagrams (below 15 mK and up to 70 mT) of YbRh_2Si_2 and $^{174}\text{YbRh}_2\text{Si}_2$;
- the expansion of the linear-in- T range of the electrical resistivity of both YbRh_2Si_2 and $^{174}\text{YbRh}_2\text{Si}_2$ to 3.5 orders of magnitude in temperature.

This list could be further expanded to include:

- the first ever electrical resistivity measurement of YbRh_2Si_2 below about 15 mK and, as a matter of fact, of any metal at ultralow temperatures, opening a new chapter in ultralow temperature physics;
- the discovery of an extreme strange metal that is not governed by the Planckian limit.

This allowed us to draw conclusions that could not have been anticipated from the Science 2016 work (Ref. 33), namely that:

- superconductivity in YbRh_2Si_2 extends to surprisingly large fields, even a factor of 3 larger than the already large estimate of the critical field (B'_{c2} , see Table II) from the upper critical field slope ($-dB_{c2}/dT|_{T_c}$, see also Table II). This is a strong indication for a mechanism boosting superconductivity at *finite* fields (i.e., the QCP) and for the unconventional nature of the superconductivity.
- the boundary of the superconducting phase of YbRh_2Si_2 is highly “non-mean-field-like” (compare with Fig. R1 right of point 2.1 above), suggesting a two-phase nature, with the high-field phase having all necessary ingredients for spin-triplet superconductivity. This will trigger further experiments to test this tentative assignment.
- in the absence of Yb nuclear moments the superconductivity is severely weakened, indicating that nuclear moments *do* play a role in *strengthening* superconductivity, but are not needed for *creating* it. This puts concise constraints on any microscopic theory for the observed superconductivity.
- the phase diagram of $^{174}\text{YbRh}_2\text{Si}_2$ clearly falls in two different phases, supporting the two-phase assignment of the superconducting phase region in YbRh_2Si_2 .
- quantum critical fluctuations, with beyond-order parameter nature as demonstrated previously, are responsible for both the extreme strange metal behavior (over 3.5 orders of magnitude in temperature) and for (at least a component of) the superconductivity of YbRh_2Si_2 , thereby pointing to a microscopic mechanism of strange metal superconductivity. We hope that our revisions (see also points 3.2-4 below) have in particular helped to underpin this last key point.

In our opinion this is a set of important (non-incremental) discoveries, and we trust that they will trigger further important work, both theoretical and experimental.

3.2 *Beyond this, the authors claim to link the superconductivity to the well-established quantum critical point and establish the superconductivity as evolving from the strange metal normal state. Undoubtedly, this would be an important step for the field. However, I do not believe that the data supports this claim. Looking at the phase diagram in fig 4, the overwhelming message is that superconductivity is strongly suppressed at the QCP, if not killed entirely. The only evidence to support superconductivity at the QCP is the slight suppression (at the few present level) of the resistance in either sample (see Figs 2a and 2b). To put this in context, the canonical unconventional superconductivity-QCP systems (e.g. CeIn_3 and High- T_c) have superconductivity nearly maximized (in terms of T_c) at the QCP. In this case it appears to me more likely that the QCP is antagonistic to superconductivity rather than the source of it. Furthermore, the lack of convincing evidence for superconductivity at the QCP does not support the claim that superconductivity evolves from the strange metal state that exhibits linear-in-temperature resistivity. In fact superconductivity is strongest, in that it has the highest T_c , where the resistivity exhibits T^2 behaviour associated with a Fermi liquid state.*

In this reasoning the Reviewer overlooks the fact that we are here using magnetic field to tune the material through its QCP, and not (chemical or external) pressure or doping as in CeIn_3 or the high- T_c 's. Because magnetic field generally suppresses superconductivity (due to the Pauli-limiting and/or orbital effect), there are now two competing trends: the suppression of T_c by magnetic field and the enhancement of T_c by approaching the QCP. We think that the shapes of the $T_c(B)$ curves shown in Fig. 4 (full and open data points) support that this is what happens here. Further details, and in particular a visualization of this effect

with simple cartoons (Fig. R2), are given in point 2.1 above, to which we kindly refer the Reviewer.

We also note that the Reviewer’s statement about the slight (“at the few percent level”) suppression of the resistivity at the QCP of YbRh_2Si_2 is incorrect, and are sorry for the unclear description that might have caused this confusion. Figure 2 (entitled “Strange metal behavior of ...”) highlights the linear-in- T behavior and thus the resistivity axis was set to just show the onset of superconductivity. The full resistivity curves are shown in Fig. 1. For YbRh_2Si_2 , at the quantum critical field of 60 mT, the resistivity has dropped to 50% at somewhat below 1 mK (Fig. 1c, green curve). Thus, T_c of YbRh_2Si_2 is clearly finite at the QCP. Moreover, the pale shadings in Fig. 4 have real meanings: they represent 90% resistance lines (thus a substantial drop of 10%). For YbRh_2Si_2 (light blue shading) there is a clear local maximum in this curve near the quantum critical field.

Changes: We have now separated the phase diagrams of YbRh_2Si_2 and $^{174}\text{YbRh}_2\text{Si}_2$ (previous Fig. 4) into two panels (new Fig. 4a,b), hoping that this increases the readability. The border of the dark blue shading—a guide-to-the-eyes for $T_c(B)$ in the original manuscript—now has more concrete meaning near the QCP: It corresponds to a linear fit to the measured high-field data points, and further underpins superconductivity extending beyond the QCP. In addition, we have expanded the description of Fig. 4 in the main text (bottom paragraph on page 4) and now explicitly remark on the “hostile” effect of magnetic fields. Finally, we have added Fig. R2 as new Fig. S3 to the Supplementary Information and refer to it in the main text. That a superconducting dome centered around a magnetic-field-induced QCP is, in general, *not* expected was already mentioned in the conclusion (3rd paragraph, page 7). Finally, to avoid the above misreading of our data (“at the few percent level”), we have improved the description of superconductivity *at* the QCP (first paragraph, page 4).

3.3 *While the idea that the superconducting state is unconventional in nature seems natural, I am looking for experimental evidence to support this statement. The authors use the fact that the rate of change of B_{c2} with temperature ($-dB_{c2}/dT$) is large as one of the main experimental results to support this idea. However, I am not aware that this is a widely used criteria for unconventionality. The problem I see is that the temperature dependence of the resistivity in the mixed state of a superconductor involves contributions from flux flow when the motion of superconducting vortices generate voltages that mimic those of normal state resistance. Without understanding the nature of this, which is typically very sample dependent due to flux pinning relating to disorder, it is difficult to draw conclusive statements about the underlying superconducting state. This kind of physics may also be relevant to the differences in the phase diagrams derived from the two types of samples, and in reconciling phase diagrams deduced from resistivity measurements in this work and those from magnetic measurements on purportedly identical samples reported earlier (Ref 33).*

Already the “historic” work on the first heavy fermion superconductor CeCu_2Si_2 reports an unusually large value of the upper critical field slope $-dB_{c2}/dT|_{T_c}$ and associates it with the high density of Cooper pairs resulting from the heavy fermion quasiparticles of the heavy fermion normal state [Rauchschwalbe et al., Phys. Rev. Lett. **49**, 1448 (1982)]. The direct correlation between $-dB_{c2}/dT|_{T_c}$ and the Sommerfeld coefficient of the specific heat (which is proportional to the electronic density of states at the Fermi level) was discovered even earlier in the chevrel phases [Ø. Fischer, Appl. Phys. **16** (1978) 1]. This understanding is still state-of-the-art.

In addition, it is the boosting of superconductivity towards the QCP—against the “hostile” effect of magnetic fields on superconductivity (see point 3.2 and 2.1 with Fig. R2 above)—that provides strong evidence for quantum critical fluctuations governing at least part of the superconductivity in YbRh_2Si_2 .

Flux-flow resistivity can be identified via a current-density-dependent electrical resistivity in the (partially) superconducting state. We have studied this effect during our initial experiments, though not having possible flux-flow effects in mind but the determination of the highest possible measurement current that does not lead to sample overheating. In Fig. R3 we show such experiments on both YbRh_2Si_2 and $^{174}\text{YbRh}_2\text{Si}_2$. The only difference between $R(T)$ [or $R(B)$] curves measured with different currents is that data measured with lower currents are noisier (the displayed data were taken with same the statistics), and that the application of too large currents leads to overheating (slightly higher resistance at low temperatures). There is, however, no indication for a flux-flow resistance.

FIG. R3: **Test measurements of current dependence of resistance.** (left) The temperature-dependent resistance data of YbRh_2Si_2 were taken at 17 mT, with excitation currents between 6 nA and 200 nA, in order to find the best current settings. (right) The field-dependent resistance data of $^{174}\text{YbRh}_2\text{Si}_2$ were taken at 1.2 mK, with two different currents. No flux-flow resistance contributions can be identified. All data presented in the manuscript were taken with currents that do not overheat the sample (10 nA at the lowest temperatures), with much longer averaging time (and improved electronics/shielding) than these preliminary measurements.

With respect to the Reviewer’s last point we note that the agreement between the present work and results from ref. 33 is quite satisfying.

Changes: We have added comments on flux flow (page 4, 2nd paragraph and Methods, Sect. E, last sentence) and improved the paragraph on the comparison with ref. 33 (page 6, 1st paragraph). In addition, we have included a new figure to the Supplementary Information (Fig. S2) that sketches two possible scenarios that are compatible with the combined results of ref. 33 and the present work, and refer to it in the text (also page 6, 1st paragraph).

3.4 *Finally, there is quite a bit of discussion towards the end of the paper concerning triplet superconductivity. Since there is absolutely no experimental evidence in this work to support any claim of this nature, this is highly speculative and I find the weight given to the discussion largely inappropriate.*

We certainly agree with the Reviewer that our evidence for spin-triplet superconductivity in YbRh_2Si_2 and $^{174}\text{YbRh}_2\text{Si}_2$ is only tentative, and we think that our concluding statement

(“Naturally, the proposal of spin-triplet superconductivity should also be scrutinized by future experiments, including probes of anisotropies and NMR investigations, which are in principle feasible at ultralow temperatures.”, see page 7, end of 2nd paragraph) should have made this sufficiently clear. Let us explain why we still think that it is interesting and appropriate to discuss this possibility. Spin-triplet superconductivity is notoriously difficult to firmly pin down, as shown maybe most prominently by the case of Sr_2RuO_4 , which had been considered as key “candidate” spin-triplet superconductor for many years but was recently concluded not to be one [Pustogow et al., *Nature* **574** (2019) 72]. It is common practice to use the comparison of the measured critical field and the Pauli limiting upper critical field as first indication for spin-triplet superconductivity. The fact that the upper critical field determined for YbRh_2Si_2 (> 60 mT, Fig. 4a) exceeds by more than a factor of four the usual Pauli limit (15 mT, see Table II) is thus a first hint that spin-triplet superconductivity could be realized.

Here, we provide two additional pieces of evidence. Firstly, in $^{174}\text{YbRh}_2\text{Si}_2$, the phase diagram clearly displays two phases. This adds evidence that a change in pairing symmetry occurs, likely also in the electronically identical compound YbRh_2Si_2 (the cartoons we draw in Fig. S3 help to see how this situation can lead to the type of phase diagrams we observe). Secondly, as discussed in the manuscript (2nd paragraph on page 7), there is theoretical support for spin-triplet pairing becoming competitive in large enough magnetic fields (ref. 39 of the manuscript).

Just as side remark, we are committed to setting up NMR experiments at ultralow temperatures—another tour-de-force effort—to scrutinize this tentative assignment. Andrej Pustogow, who was just appointed at our institute, will join this effort.

To summarize, I do not support publication of the current manuscript in Nature Communications because I do not find that the data presented supports the broad conclusions that the authors need to make to bring the work to necessary level of impact for this journal. Moreover, the data itself is insufficient to warrant publication because it is supplemental to the original discovery of superconductivity in this material, which was reported some years earlier.

We hope that the Reviewer is satisfied with our replies, and finds that the changes to the manuscript have clarified both the level of achievement beyond the Science-2016 work and the connection between the presented data and conclusions drawn from them.

Reviewers' Comments:

Reviewer #3:

Remarks to the Author:

This is my third review of the paper, "Deciphering high-temperature superconductivity with ultralow temperatures" by Nguyen et al. In my previous review, I argued that a possible interpretation of the experimental data is that the fluctuations associated with the QCP are antagonistic to superconductivity, which is why the superconducting phase has a critical field that is either essentially the same as the quantum critical magnetic field (Yb – natural abundance) or well below it (Yb – isotopically pure). At that point, the authors had only considered the possibility that superconductivity results from the same fluctuations that arise from the quantum critical point. In turn, this allowed for a broader interpretation of the results in the context high-T_c superconductivity.

In response to this, the authors have not provided any new evidence or arguments. They highlight the unusual magnetic field dependence of the isothermal resistivity measurements (Fig. 3a, b) as the major observation to promote a connection between superconductivity and quantum criticality. While it is plausible and consistent with the data to point out that potentially, in the absence of quantum fluctuations, the superconductivity might be suppressed more quickly by magnetic field, it is (in my opinion) a stretch to say this is the only reasonable explanation. There are many unanswered questions about the superconducting state in this material; what is the pairing mechanism?, is this single band or multi-band superconductivity?, is it multi-phase? Nor do we know how homogeneous the superconductivity is; surface or bulk?, what is the interplay between superconductivity and magnetism?

There is no doubt that this discussion is critical to the manuscript because it is necessary to go beyond the simple story the data exposes in Fig 4 (which is: in the standard sample superconductivity dies at the quantum critical point (is this a co-incidence?), while in the isotopically pure sample, there is no superconductivity at the QCP). However, with the current experimental evidence, I believe that we cannot truly know whether quantum fluctuations promote or are antagonistic to superconductivity.

With so many uncertainties, I do not see a clear path to the conclusion that "critical fermionic modes.... appear to mediate the strange-metal superconductivity in YbRh₂Si₂." and with it the title of "Deciphering high-temperature superconductivity with ultra-low temperatures".

In the end, I would not recommend publication of the current manuscript because in my opinion, it does not provide a balanced interpretation of the data.

Point-by-point reply

Reviewer 3 – 3rd report

This is my third review of the paper, “Deciphering high-temperature superconductivity with ultralow temperatures” by Nguyen et al.

We indeed thank the Reviewer for taking once again the time to comment on the resubmitted documents. We address his/her remaining points below, and also append the previous round’s point-by-point reply with Reviewer 3 at the end of this document for convenience.

A *In my previous review, I argued that a possible interpretation of the experimental data is that the fluctuations associated with the QCP are antagonistic to superconductivity, which is why the superconducting phase has a critical field that is either essentially the same as the quantum critical magnetic field (Yb – natural abundance) or well below it (Yb – isotopically pure). At that point, the authors had only considered the possibility that superconductivity results from the same fluctuations that arise from the quantum critical point. In turn, this allowed for a broader interpretation of the results in the context high- T_c superconductivity. In response to this, the authors have not provided any new evidence or arguments. They highlight the unusual magnetic field dependence of the isothermal resistivity measurements (Fig. 3a, b) as the major observation to promote a connection between superconductivity and quantum criticality. While it is plausible and consistent with the data to point out that potentially, in the absence of quantum fluctuations, the superconductivity might be suppressed more quickly by magnetic field, it is (in my opinion) a stretch to say this is the only reasonable explanation.*

We are disappointed to see that, in spite of all our efforts to explain the dual role magnetic field plays (see, e.g., bottom two paragraphs on page 5 and Fig. N1), this Reviewer continues to doubt the (in our opinion overwhelming) evidence for quantum criticality stabilizing at least part of the superconductivity in YbRh_2Si_2 . Following the advice of the Editor, we thus now summarize this evidence in an extra paragraph in the manuscript, which reads (lines 173-184):

We start by recapitulating our results that make the BCS mechanism extremely unlikely: (i) Superconductivity in YbRh_2Si_2 condenses out of an extreme strange metal state, with linear-in-temperature resistivity right down to the onset of superconductivity (Fig. 2a); (ii) the upper critical field slope (Table II) as well as the directly measured critical field (Fig. 4a) strongly overshoot both the Pauli and the orbital limiting fields; (iii) the low-temperature resistivity isotherms exhibit a two-step transition (Fig. 3a), evidencing that one component is much less field-sensitive than the other; (iv) the superconducting phase boundary deviates strongly from a mean-field shape (Figs. 3a and 4a), evidencing that the field boosts (at least part of) the superconductivity against the general trend of field suppression; (v) superconductivity is strongly suppressed by substituting the natural abundance Yb (of atomic mass 173.04) by ^{174}Yb , though the isotope effect in a BCS picture would have a minimal effect (a reduction of T_c by 0.1%). It is thus natural to assume that quantum critical fluctuations are involved in stabilizing (at least the high-field part of) the superconductivity in YbRh_2Si_2 .

B *There are many unanswered questions about the superconducting state in this material; what is the pairing mechanism?, is this single band or multi-band superconductivity?, is it multi-phase? Nor do we know how homogeneous the superconductivity is; surface or bulk?, what is the interplay between superconductivity and magnetism?*

There is no doubt that this discussion is critical to the manuscript because it is necessary to go beyond the simple story the data exposes in Fig 4 (which is: in the standard sample superconductivity dies at the quantum critical point (is this a co-incidence?), while in the isotopically pure sample, there is no superconductivity at the QCP). However, with the current experimental evidence, I believe that we cannot truly know whether quantum fluctuations promote or are antagonistic to superconductivity.

We have never claimed that all aspects of the superconductivity in YbRh_2Si_2 and $^{174}\text{YbRh}_2\text{Si}_2$ have been settled by the present study. In fact, there were two paragraphs in the manuscript explicitly alluding to the need for further work, one about clarifying the nature and arrangement of different superconducting phases (lines 155-158):

“This should be clarified by future magnetization/susceptibility measurements in lower fields (below the background field of 0.012 mT reached in ref. 33, which appears to be well above the lower critical field of the B phase), ideally on powdered samples to better assess the Meissner volume of the B phase.”

and one about testing the proposal of spin-triplet superconductivity (lines 210-212):

“Naturally, the proposal of spin-triplet superconductivity should also be scrutinized by future experiments, including probes of anisotropies and NMR investigations, which are in principle feasible at ultralow temperatures.”

Nevertheless, we have now added an additional, more general paragraph in the concluding part (lines 220-223):

“Future experiments, ideally in conjunction with *ab initio*-based theoretical work, shall ascertain this assignment, disentangle the different superconducting phases, determine the symmetry of the order parameter, clarify further important details such as the single vs multiband nature of the superconductivity, and even explore the possibility of exotic surface phases.”

C *With so many uncertainties, I do not see a clear path to the conclusion that “critical fermionic modes ... appear to mediate the strange-metal superconductivity in YbRh_2Si_2 .” and with it the title of “Deciphering high-temperature superconductivity with ultra-low temperatures”.*

We have softened both statements. The new title reads “**Superconductivity in an extreme strange metal**”, the last two sentences in the abstract are revised to “We **propose** that the Cooper pairing is mediated by fermionic modes associated with a recently evidenced dynamical charge localization–delocalization transition [12], **a mechanism that** may well be pertinent also in other strange metal **superconductors**.”, and the concluding sentence is amended to “Our results **thus point to the exciting possibility that a dynamical electron localization–delocalization transition may mediate strange-metal unconventional superconductivity in a broad range of materials classes.**”

In the end, I would not recommend publication of the current manuscript because in my opinion, it does not provide a balanced interpretation of the data.

We hope that the Reviewer is satisfied by these changes and now finds our work ready to be seen by the scientific community, for further scrutiny.

Reviewer 3 – 2nd report

This is my report on the second revised version of the manuscript, “Deciphering high-temperature superconductivity with ultralow temperatures” and the accompanying, “point by point reply” document that is a response to the first round of referees comments.

I agree with the majority of the responses by the authors to the comments specifically to reviewer 3. Furthermore, incorporation of many of the reviewers comments means that the revised manuscript is improved in terms of clarity in comparison to initial submission.

We are also grateful to Reviewer 3 for taking the time to evaluate the revised manuscript and our replies to his/her previous comments. We are glad to see that he/she is satisfied by most of our responses and concludes (see below) that “there is still a strong case for publication in Nature Communications”. Of course we gladly address his/her remaining concern. For clarity, we label the paragraphs of the present report (pasted below) by **a**, **b**, **c**... and refer to points of the Reviewer’s previous report (appended at the end) by **3.1**, **3.2**, **3.2**....

a, *However, I still find that the data presented does not unambiguously support the larger claims by the authors that would elevate the work beyond what is still an interesting report on low temperature transport properties of YbRh₂Si₂ in its natural and isotopically pure forms.*

b, *The key area of concern for me remains the establishment of the relationship between the linear in temperature resistivity and superconductivity in YbRh₂Si₂. In the fourth paragraph of the manuscript, the authors state that their report “establishes the connection between electron localization-delocalization-derived [48] strange metal behavior and unconventional superconductivity, discussed previously for several [49] other materials [21-24], to an unprecedented level of confidence, ...”. This is a bold claim and requires considerable experimental scrutiny.*

c, *The data at hand to support this claim are provided in Figure 1. They are also the topic of point 3.2 in the rebuttal document, where the authors write, “The full resistivity curves are shown in Fig. 1. For YbRh₂Si₂, at the quantum critical field of 60mT, the resistivity has dropped to 50% at somewhat below 1mK (Fig. 1c, green curve). Thus, T_c of YbRh₂Si₂ is clearly finite at the QCP. Moreover, the pale shadings in Fig. 4 have real meanings: they represent 90% resistance lines (thus a substantial drop of 10%). For YbRh₂Si₂ (light blue shading) there is a clear local maximum in this curve near the quantum critical field.”*

d, *An alternative description of the data would be the following. At the quantum critical field (60 mT), the resistivity of the natural sample of YbRh₂Si₂ at the lowest available measurement temperatures has fallen to 50% of the normal state value, indicating that a considerable amount of the sample remains in the normal state. For the isotopically pure sample, the resistivity has dropped by only a few percent, indicating that the vast majority (> 90%) of the sample is still in the normal state. Implicit here is the difficulty of knowing exactly how much of the sample is truly superconducting on the basis of resistivity measurements alone. Furthermore, the trajectory of the temperature dependences (at 60 mT) on the basis of the current data set does not lead to zero resistivity at any finite temperature.*

e, *The attribution of T_c to when the resistivity falls to 50% of its normal state value is a widely used definition, but this is usually in a situation where the full transition is available. To use it as a basis for defining a superconducting transition temperature in the absence of the full transition is less obvious since a definition that is arbitrarily lower, for example 40%, does not allow one to define a transition temperature in either sample at the quantum critical field.*

f, *The authors quite rightly state that the application of a magnetic field to YbRh₂Si₂ has two*

effects. On the one hand, it is the tuning parameter necessary for accessing the quantum critical point, while on the other it is conventionally antagonistic to superconductivity. This is simply an unavoidable (and in this case unfortunate) effect of nature, which makes superconductivity and quantum criticality difficult to study simultaneously in this material. So, while YbRh_2Si_2 may avoid the complexity of phase diagrams that hamper the deciphering of the relationship between superconductivity and the strange-metal normal state, it has its own unique issues concerning the dual effects of magnetic field.

g, For this reason, the authors are forced to infer that superconductivity evolves out of the strange metal state by bolstering the limitations of the data with models in which they compensate for the effect of the magnetic field on superconductivity. This is markedly different from the data itself providing an “unprecedented level of confidence” on this issue and the assuredness of the wide-reaching claims, which include the title of the manuscript itself.

h, In the end, it is not whether I believe that there is a relationship between the strange metal state and superconductivity in this material, but that the authors need to improve the clarity of what is unambiguous according to the data they present and what is an assumption. The bottom line here is that, on the basis of the data, the superconductivity in this material is strongest (in terms of highest T_c) in the Fermi-liquid (T_2) regime at zero magnetic field. It is either non-existent or only partially present at the lowest possible measurement temperatures at the quantum critical magnetic field. It is also an intriguing observation that in the natural sample of YbRh_2Si_2 the disappearance of superconductivity is at almost the exact field as the quantum critical point. Examining this issue might also require better knowledge of the precise value of the quantum critical field itself, about which the current manuscript is notably lacking. To balance the appraisal of the data, I think it is reasonable to expect the authors to address this perspective.

i, In summary, my recommendation is that current version of the manuscript is unsuitable for publication in its current form. The authors must clearly distinguish between what is factual according to experimental data and what conclusions are drawn based on reasonable assumptions. Of course, my expectation is that this will somewhat restricts the appeal of the manuscript, but given the extended list of experimental achievements provided in response 3.1, there is still a strong case for publication in *Nature Communications* (apologies for the reference to *Nature Physics* in my first report).

In the first two paragraphs (**a** and **b**) the Reviewer expresses his/her remaining concern, and details it further in the following paragraphs (**c** - **h**).

We thank him/her for now summarizing our temperature-dependent electrical resistivity data from Fig. 1 correctly (**c**, see **3.2** for the previous oversight of the relevant panels). Note that important additional data, not mentioned by the Reviewer but equally used to construct the phase diagrams of Fig. 4, are the isothermal magnetic-field curves in Fig. 3 a, b, as well as the color-coded phase diagrams in Fig. 3 c, d (both discussed further below).

As to the discussion in **d** and **e**, we are not sure what the Reviewer wants to implicate. To us, there cannot be any reasonable doubt about the presence of superconductivity in our data. Then, the precise definition of T_c boils down to being a rather technical issue. Any superconducting transition has a finite width and, as the Reviewer acknowledges in **e**, to define T_c at the half-height is a widely used criterion. We also note that a visual extrapolation on linear scales to judge whether or not the resistivity will fall to zero at lower temperatures may not be reliable for the natural temperature scale is logarithmic.

For our conclusions, however, another point is more important than the precise T_c definition.

FIG. N1: **Replotted from FIG. S3 b.** Cartoon of the (mean-field like pair-breaking) magnetic-field effect on a superconducting dome around a quantum critical point (QCP).

As seen from Fig. 3 a, b, below a certain temperature, the resistivity vs field isotherms show a “two-step” structure (that is not due to flux flow, as discussed in **3.3**). The lower temperature part of it is boosted by the magnetic field, *against* the general trend of field suppression that the Reviewer now acknowledges (**f**, and **3.2** for the previous oversight of the field suppression effect). The boosting effect is also seen in the totally unbiased color-coded plots in Fig. 3 c, d. We see no other reasonable explanation for it than that the proximity to the quantum critical point is responsible for this superconductivity. This is, to us, strong evidence for the intimate connection of quantum criticality and at least this high-field superconducting phase of YbRh_2Si_2 .

Stepping back, one should acknowledge that YbRh_2Si_2 is a material that, already natively (in zero field), is situated in extreme vicinity to a quantum critical point. The Néel temperature is only 70 mK and the quantum critical field is only 60 mT. Quantum critical behavior that emerges from the QCP (at 60 mT and 0 K) expands quickly over wide field ranges as temperature is increased; already at 90 mK, the quantum critical fan [orange “tornado” of Fig. 1a of Custers et al., Nature 424 (2003) 524] extends to zero field. As such it seems likely that even the low-field superconducting phase is due to quantum critical fluctuations and not by coincidence occurring in such vicinity to a QCP.

We had tried to illustrate this situation in the last round by adding the cartoons of Fig. S3 to the Supplementary Information. We paste panel b of this figure below because it helps to further illustrate the Reviewer’s point. The fat red curve is a hypothetical $T_c(B)$ curve where any “hostile” effect of the magnetic field on superconductivity (from Pauli or orbital limiting) is imagined to be absent. For the successive curves (from red to brown), an increasingly strong “hostile” field effect is considered (using a simple mean-field-type suppression of T_c by the magnetic field B , which increases from zero along the tuning parameter axis). The purple curve is roughly what we observe in YbRh_2Si_2 . The physics, however, would still be the same if we had observed a further suppressed dome (brownish curves).

On the discussion in **f** we only partially agree. Whereas magnetic field as control parameter to reach a QCP has the disadvantage of superimposing a pair-breaking effect, it helps to (i) distinguish phases of different symmetry (singlet vs triplet pairing) via their different field sensitivity. Furthermore, *because* superconductivity is weakened, (ii) normal state non-Fermi liquid behavior can be accessed down to lower temperatures (establishing the record span of 3.5 orders of magnitude of linear-in- T range, Fig. 2 c). Finally, (iii) magnetic field is a continuous and clean control parameter that is available even at ultralow temperatures. As such, we consider the setting of YbRh_2Si_2 rather fortunate.

As to point **g** we hope that the above discussion clarifies that we are not “bolstering the limitations of the data with models” but instead are using a transparent and standard T_c definition as well as fully unbiased color-coded plots. (Fits with the simple two-phase model of the Supplementary Information were only done for $^{174}\text{YbRh}_2\text{Si}_2$; they only affect the region between the two phases.)

Finally, we address the Reviewer’s comment in **h** that a “better knowledge of the precise value of the quantum critical field itself” would be needed. We are sorry for not having explicated this in the caption of Fig. 4. The most precise determination of the quantum critical field of YbRh_2Si_2 for the direction we study here ($B \perp c$) comes from Refs. 28,29. For convenience, we replot the relevant panels below. In fact, we were conservative in assuming a critical field value of 60 mT. The right panel in Fig. N2 may suggest an even smaller value. We have now amended the caption of our Fig. 4 accordingly.

FIG. N2: **Higher-temperature phase diagrams of YbRh_2Si_2 from the literature.** Both locate the QCP at 0.06 mT at maximum. Left panel from Ref. 28, right panel from Ref. 29. These single crystals, as well as ours, were all grown by Cornelius Krellner by the same technique (Ref. 32).

We also gladly took up his/her recommendation (in **i**) to screen the manuscript for cautious wording and have amended the formulations at several places. These changes, as well as the amendment of the caption of Fig. 4 discussed above, are printed in red in an extra copy of the revised manuscript.

We hope that Reviewer 3 is satisfied with our explanations and the changes to the manuscript, and can now recommend publication of the manuscript.